# Maintenance of pluripotency-like signature in the entire ectoderm leads to neural crest stem cell potential

Ceren Pajanoja [1,2], Jenny Hsin [1], Bradley Olinger[1], Andrew Schiffmacher [1,5], Rita Yazejian [1], Shaun Abrams [1], Arvydas Dapkunas [2], Zarin Zainul[1], Andrew D. Doyle[3], Daniel Martin [4] & Laura Kerosuo [1,2] ✉

The ability of the pluripotent epiblast to contribute progeny to all three germ layers is thought to be lost after gastrulation. The later-forming neural crest (NC) rises from ectoderm and it remains poorly understood how its exceptionally high stem-cell potential to generate mesodermal- and endodermal-like derivatives is obtained. Here, we monitor transcriptional changes from gastrulation to neurulation using single-cell-Multiplex-Spatial-Transcriptomics (scMST) complemented with RNA-sequencing. We show maintenance of pluripotency-like signature (*Nanog*, *Oct4/PouV*, *Klf4*-positive) in undecided pan-ectodermal stem-cells spanning the entire ectoderm late during neurulation with ectodermal patterning completed only at the end of neurulation when the pluripotency-like signature becomes restricted to NC, challenging our understanding of gastrulation. Furthermore, broad ectodermal pluripotency-like signature is found at multiple axial levels unrelated to the NC lineage the cells later commit to, suggesting a general role in stemness enhancement and proposing a mechanism by which the NC acquires its ability to form derivatives beyond "ectodermal-capacity" in chick and mouse embryos.

Pluripotent epiblast stem cells, with the potential to become every cell type in the body, become restricted to germ layer-specific fates during gastrulation. Morphogen gradients polarize the body axes leading to proper tissue patterning. Medial-lateral patterning of the ectoderm germ layer results in formation of the medial future central nervous system (CNS) domain flanked by the lateral non-neural ectoderm (NNE) domains that become the skin[1] (Fig. 1a). The neural plate border (NPB) between these two domains gives rise to the neural crest (NC) and cranial placodes. The NC represents an exception to the idea of germ layer restriction because in addition to giving rise to ectodermal-like derivatives such as the peripheral nervous system and pigment cells[2,3], it also differentiates to facial bones and cartilage, smooth muscle, adipocytes[4–6], and endocrine cells such as chromaffin cells in the adrenal medulla and the calcitonin secreting cells in the ultimobranchial body[3,7] – non-ectodermal-like cell types that would typically arise from the mesodermal and endodermal germ layers[8]. Although the steps of neural crest development are well established and conserved across species from induction at the NPB to specification at the dorsal neural tube at the premigratory stage, further followed by epithelial-to-mesenchymal transition that allows transition into a

[1]National Institute of Dental and Craniofacial Research, Intramural Research Program, Neural Crest Development and Disease Unit, National Institutes of Health, Bethesda, MD, USA. [2]Department of Biochemistry and Developmental Biology, Faculty of Medicine, University of Helsinki, Helsinki, Finland. [3]National Institute of Dental and Craniofacial Research, Intramural Research Program, NIDCR Imaging Core, National Institutes of Health, Bethesda, MD, USA. [4]National Institute of Dental and Craniofacial Research, Intramural Research Program, Genomics and Computational Biology Core, National Institutes of Health, Bethesda, MD, USA. [5]Present address: Department of Animal and Avian Sciences, University of Maryland, College Park, MD, USA. ✉e-mail: laura.kerosuo@nih.gov

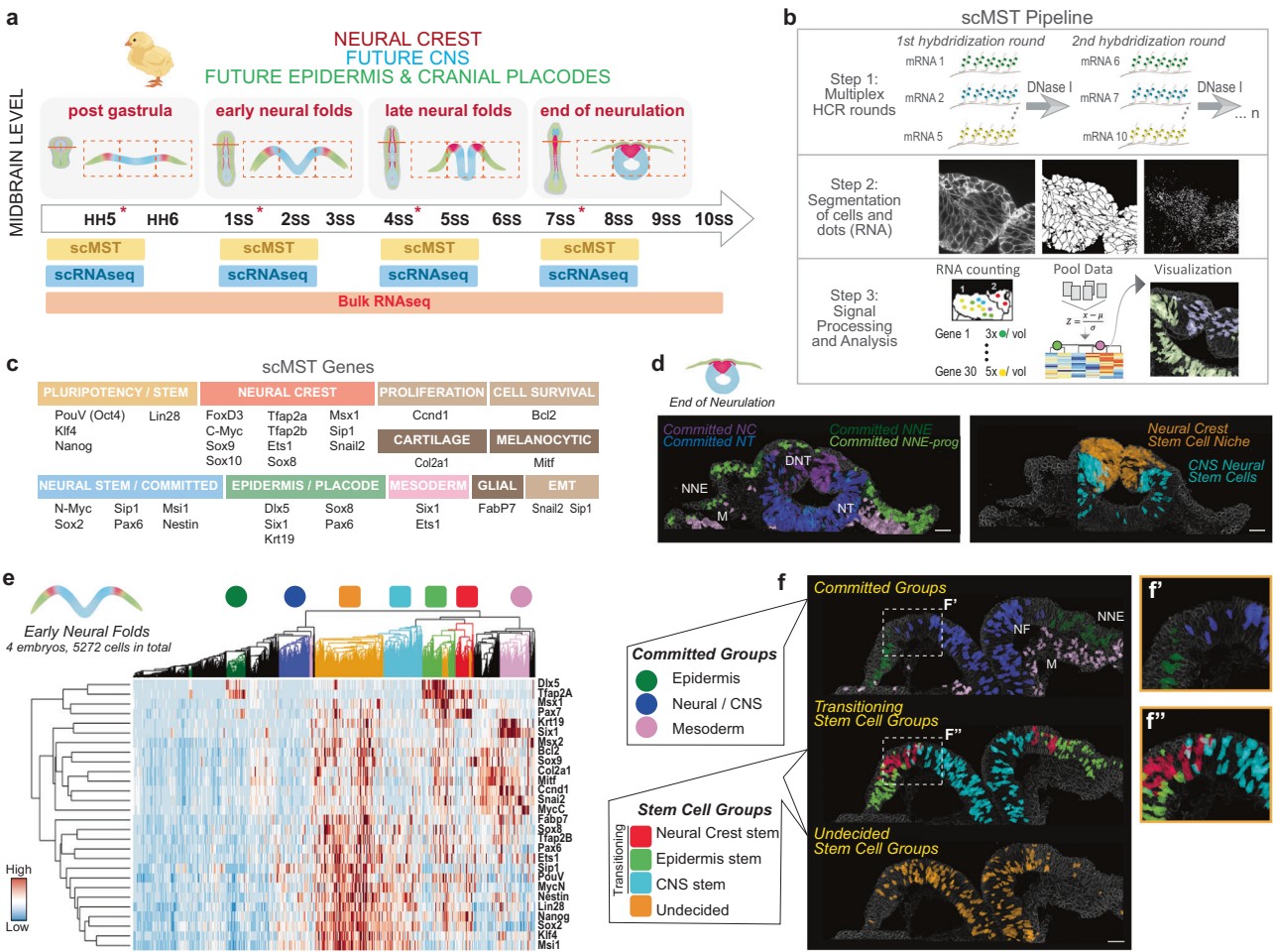

**Fig. 1 | Multiplex Single Cell Spatial Transcriptomics reveals pluripotent-like cells broadly throughout the neurulating ectoderm. a** Experimental plan and a schematic demonstrating the rise of the neural folds and how the ectoderm is patterned into three domains during neurulation. Dashed squares demonstrate fields of views imaged for scMST. All twelve stages were used for bulkRNAseq and asterisks highlight the four stages that were used for scMST and scRNAseq. The Chick (yellow) image was created using BioRender.com, HH Hamburger Hamilton stage, SS Somite Stage. **b** The scMST pipeline demonstrates serial hybridization rounds, 3D segmentation of cells and how individual RNA transcripts (white dots) are quantified and analyzed per cell. Then, pooled data of transcript counts in individual cells is pooled into a heatmap followed by back-mapping of cells in the subclusters to visualize their location in the original tissue sections by using pseudo-coloring. **c** List of genes chosen for scMST. **d** Pseudo-colored scMST subpopulations at the end of neurulation consists of committed cells as well as separate domains of neural crest and neural stem cells, respectively. *n* = 4 biological replicates. Scale bar = 30 μm. **e** scMST results from early neural fold stage highlight broad existence of cells with a pluripotency-like signature throughout the developing ectoderm. Based on unbiased hierarchical clustering, the heatmap shows transcriptionally defined subpopulations labeled with a color. Circle symbols indicate committed cell groups and square symbols indicate stem cell groups. **f** Each single cell is mapped back into the original position in the embryo image and pseudo-colored according to their representative subpopulation as labeled in the heatmap in e. *n* = 4 biological replicates. Scale bar = 30 μm. The stem cell groups co-express pluripotency genes together with ectodermal domain markers. Cells in the undecided stem cell population (orange) are situated across the entire ectoderm and demonstrate highest stem cell potential as the future ectodermal domain they will commit to can not be predicted from the transcriptional profile. **f'** Higher magnification of committed subpopulations revels that the cells of different groups do not overlap spatially, whereas **f"**, transitioning stem cell groups show spatial overlap. Source data for scMST are provided as a Source Data file. NNE non neural ectoderm, DNT dorsal neural tube, NT neural tube, NF neural fold, M mesoderm.

migratory stage[8–16] (Fig. 1a), how the NC establishes this exceptionally high pluripotent-like stem cell state remains poorly understood.

Two contradicting hypotheses have been recently proposed to explain this intriguing conundrum. One model suggests the exceptionally high stem cell potential of the NC is retained from blastula stage (the Xenopus animal cap ectoderm, equivalent of the mammalian epiblast) mainly based on continuos expression of key NC genes that were thought to promote pluripotency[17]. A more recent study used Wnt1Cre and Oct4 reporter mouse lines to conclude that rather than maintaining pluripotency, the high level of NC stemness is enabled by the unique ability of the NC to re-activate the pluripotency-like signature during the neurulation process at late neural fold stages well after gastrulation is completed[18]. Neither of these hypotheses are supported by scRNAseq analyses of whole frog or zebrafish embryos[19,20]. While the transcriptomic approaches found no sign of a distinct pluripotent expression program that persists from blastula to give rise to the neural crest, no "later" neural crest stem cell population with a reactivated pluripotency-like signature was identified either, but rather a conventional differentiation pathway of neural crest formation from intermediate neuroectodermal cell clusters was proposed[19,21]. Although neuro-ectodermal expression of the pluripotency factors Nanog and Oct4 has been reported in the neurulating mouse, frog and chicken embryos[18,22–26], the scRNAseq analysis data set, possibly partially due to technical limitation caused by insufficient resolution from the whole embryo samples, could not detect separate clusters of pluripotent cells. Importantly, expression of pluripotency genes was reported across multiple ectodermal populations[19], which may partially be a reflection of how the clustering algorithms select for

strong differences between cell populations, causing stem cells, which tend to express fluctuating, lower levels of genes from multiple future lineage options[27], to be thrown to clusters of several different committed lineages based on their natural, dynamic heterogenic gene expression profiles.

We recently used the previous version of the method we developed, single cell Multiplex Spatial Transcriptomics, scMST (Fig. 1b), to investigate NC stemness at the end of neurulation in the chick dorsal neural tube[28]. The results revealed a subset of NC cells alongside the dorsal midline that co-express *Nanog*, *PouV (Pou5f3)/Oct4* and *Klf4* transcription factors[26], which are core components of a gene module that drives pluripotency in embryonic stem cells[29–31]. We defined this domain as a transient NC stem cell niche[26]. Recent studies have since confirmed the expression of pluripotency genes in the NC also in mouse and frog embryos[18,22]. Furthermore, ectopic expression studies have shown that Oct4 reactivates Nanog and regains a pluripotency status in the pre-somitogenesis stage mouse embryo ectoderm[25], and ectopic Ventx2, the frog homolog of Nanog, in the ectoderm is essential for enabling the ectomesenchymal potential of the neural crest[22]. Taken together, while multiple reports provide evidence on the expression of pluripotency-related genes in the developing ectoderm, to this day, a systematic, multiplexed, high resolution, single-cell level spatiotemporal transcriptional examination of the stem cell profiles during gastrula-to-neurula stages has not been performed, which will provide insight into the timing when the neural crest starts to express pluripotency genes and whether this is maintained or reactivated. Understanding how NC forms is highly clinically relevant: ~25% of all birth defects comprise of neurocristopathies, and the demand for using NC cells for regerative purposes is increasing. Furthermore, understanding of whether climbing upward the Waddington's epigenetic landscape[32] truly is part of embryogenesis will provide important clarity to the comprehension of pluripotency regulation in normal as well as malignant tissue growth.

To this end, we investigated temporal changes in expression of pluripotency genes to gain insights into how the developing ectoderm can acquire a domain with high, pluripotency-like features so late in development. Multiple high-resolution approaches, including scMST, single cell, and bulk RNA sequencing (scRNAseq, bulk RNAseq) were utilized to monitor gene expression across a broad series of developmental stages for a comprehensive analysis at the midbrain level (Fig. 1a), as this region of the NC gives rise to a diverse set of cell types including the craniofacial skeleton.

## Results

### Spatial transcriptomics identifies ectodermal pluripotent-like cells

30 genes that reflect all three ectodermal domains (NNE, NC, CNS) as well as pluripotency factors, were chosen for our scMST analysis (Fig. 1c). Due to the small contribution of the midbrain region to placodal development, placodes were not addressed in the analysis. To analyze the entire ectoderm, the computational pipeline for scMST gene expression analysis was optimized for both multiple developmental stages and several fields of views captured per section (Fig. 1a). Individual cells of each developmental stage (collected from four embryos per stage and 1–2 midbrain sections per embryo, consisting of a total pooled number of 4866–8253 cells per stage) were pooled into a heatmap and hierarchically clustered into transcriptionally distinct subpopulations. Based on their transcriptional profile, each subpopulation was annotated and assigned a color. Then each cell was visualized by mapping it back to the original embryo image by pseudo-coloring the cells on the sections with the same color as their respective subpopulation in the heatmap, herein referred to as spatial back-mapping. Furthermore, the clusters were also visualized in a UMAP to demonstrate their transcriptional differences (Fig. 1b–f, Supplementary Figs. 1–3, Source Data). Note that the supplementary

images show backmapping to several embryos per stage demonstrating reproducibility of the spatial patterns. We analyzed pluripotency gene expression at four different stages from gastrulation to the end of neurulation, starting with the oldest stage. Cells in the dorsal neural tube (DNT, 7 somites) at the end of neurulation, contained two stem cell populations (Fig. 1d and Supplementary Fig. 2a-a''): (1) NC stem cells (orange), which co-express NC markers, but also genes associated with the neural and epidermal domains, alongside the pluripotency genes *Nanog, PouV/Oct4, Klf4* together with *Lin28*, suggesting a stem cell state that keeps options open for multiple cell fates, and (2) neural stem cells (light blue) that have a predominantly CNS profile, but also co-expressed pluripotency genes and a low level of NC genes. We speculate that this unique neural stem cell gene expression profile may provide plasticity to the DNT as it, right after the stage we analyzed as our last timepoint, will go through major reconstruction to form the roof plate after the NC has emigrated from the neural epithelium[33]. The NC stem cells also co-express neural, glial, pigment and mesenchymal genes further suggesting that at this premigratory stage they remain unspecified regarding their future NC lineage. Importantly, the DNT also consists of committed cells, including NC cells that do not have a stem cell profile and only express NC genes (Fig. 1d and Supplementary Fig. 2a-a''), in line with our previous findings[26].

To understand how NC stem cell potential arises between gastrulation and neurulation, we next analyzed the ectoderm at three different developmental stages that encompass neural fold elevation (Fig. 1a). At late neural fold stage (4 somites), we detected committed cells (circle symbols) of all three domains in the correct spatial positions, which notably, do not show spatial overlap (Supplementary Fig. 2b–b''). Unexpectedly, at this relatively late post-gastrulation stage when the ectodermal domains are thought to be spatially restricted and fully committed, we discovered stem cell subgroups that co-expressed the pluripotency genes (square symbols). Based on their gene expression profile, we divided the stem cells into two groups. First, we detected transitioning stem cells in all three respective domains, which already strongly expressed genes of one respective domain: both CNS and the NNE stem cells still also expressed some NPB markers, whereas NPB stem cells expressed a broad array of genes including markers of both neighboring domains. Importantly, the cells in these subpopulations also spatially overlapped with each other in the ectoderm reflecting an intermediate, transitioning status (Supplementary Fig. 2b''). Finally, we identified an additional stem cell group with an uncommitted transcriptional profile, of which the future ectodermal domain could not be predicted based on the heatmap and that spatially was mostly located in the NC and CNS domains (orange cluster in Supplementary Fig. 2b''). Combined, these results show that the ectoderm is not fully committed to the three domains at the late neural fold stage and suggest that cells in each subdomain of the ectoderm have much higher plasticity than previously known. The transition from undecided to committed polulations via the transitioning transcriptional states was also recapitulated in the UMAP (Fig. 2b')

Next, we analysed embryos at the early neural fold stage (1 somite). Intriguingly, we detected evidence for even broader pluripotency-like signature across the entire ectoderm: while some committed cells were found at their anticipated locations in the future epidermis and CNS, the majority of the cells met the criteria of a stem cell as evaluated by co-expression of the pluripotency genes (Fig. 1e, f'' and Supplementary Fig. 3a-a'). Similar to the late neural fold stage data, we detected subpopulations of transitioning stem cells (light blue CNS; red NPB; light green NNE) that, based on the strongest expression of genes related to one of the ectodermal domains, were already transitioning to a fate of one of the ectodermal domains while still co-expressing pluripotency markers as well as markers of the neighboring domains. Spatial back-mapping showed that the domains of these transitioning stem cells overlapped with each other with intermingling

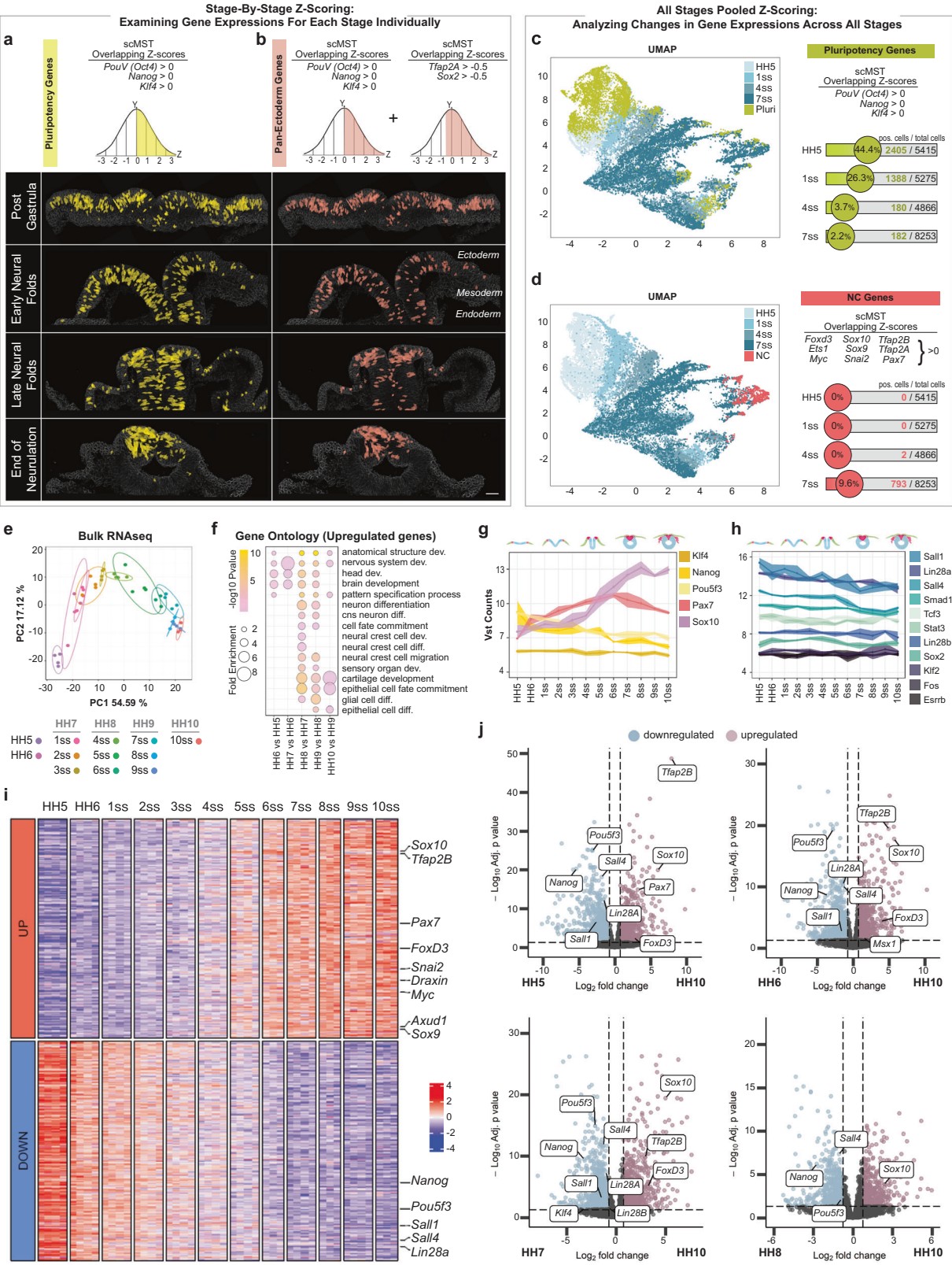

cells appearing between domains, whereas the committed groups were clearly separated (Fig. 1f–f"). Most importantly, this earlier stage contained a large stem cell population with a pluripotency-like signature (orange), which based on the co-expression of all respective ectodermal domain markers consisted of cells that were not committed to a specific domain fate. Spatial back-mapping of these undecided stem cells into their respective original tissue locations

showed that they spanned all three ectodermal domains. Visualization of the transcriptional relationships of these subgroups in a UMAP also reflected the ongoing gradual commitment process in the ectoderm (Fig. 1f and Supplementary Fig. 3a, a'). Finally, at the earliest stage examined immediately after gastrulation (Hamburger Hamilton Stage 5, HH5), we found that the majority of ectodermal cells clustered into the undecided stem cell group (orange) that spanned the entire

**Fig. 2 | Pluripotency-like signature is maintained from gastrulation to the end of neurulation. a** Cells with a pluripotency-like signature (co-expressing *Nanog, PouV/Oct4 and Klf4*) with z-scores above the mean for all 3 genes were selected using scMST. Z-scores were calculated at each developmental stage separately. Their visualization (yellow) highlights maintenance of cells with a pluripotency-like signature from gastrula stage in the entire ectoderm until the end of neurulation when they are restricted to the dorsal neural tube. **b** Similarly, pan-ectodermal stem cells co-express pluripotency genes together with the respective non-neural ectoderm (*Tfap2A*) and CNS neural (*Sox2*) markers (pink). *n* = 4 biological replicates for each stage. Scale bar = 30 μm. **c** Gene expression was z-scored across all stages of scMST samples to compare representation of cells with a pluripotency-like signature during ectoderm development, as marked with light green on a UMAP. The number and percentage of selected cells from each stage are displayed on the plot. **d** Similarly, cells co-expressing nine NC specifier genes with z-scores above the mean were selected and marked with red on the UMAP plot, which demonstrates the onset of NC specification at the end of neurulation. **e** Principal component analysis shows separation of bulk RNAseq data collected from NC domains according to developmental stage. **f** Differentially expressed gene ontology term representations from bulk RNAseq data reflect NC development. The pooling of somite stages (SS) into Hamburger Hamilton (HH) stages is shown in the list (**e**). Fold enrichments obtained via statistical overrepresentation test. *P*-values calculated using Fisher's Exact test, adjusted with Benjamini-Hochberg FDR. **g, h** VST (variance stabilizing transformation) normalized counts plotted on line plots with shaded error ribbons across all stages, indicating a continuation of pluripotency gene expression with no gaps from gastrula to end of neurulation. *n* = 4 biological replicates for samples HH5-HH6, 1ss-5ss, 7ss and 9ss. *n* = 3 biological replicates for samples 6ss, 8ss and 10ss. Error bars = SEM. **i** Heatmap displays top 250 up- and downregulated genes (*p*-adj < 0.05, LFC 0.75) in the neural crest domain across twelve developmental stages, analyzed using DESeq2 LRT (Likelihood Ratio Test). Pluripotency genes are gradually downregulated while neural crest genes are upregulated with an intermediate co-expression period during mid- and end of neurulation stages. For full list see Supplementary Data 1, 2. **j** Volcano plots of the Wald test show significantly upregulated (red) and downregulated (blue) genes at HH10 compared to earlier NC stages (*p*-adj < 0.05, LFC 0.75).

ectoderm, and of which the future ectodermal domain was not predictable based on the transcriptional profile presented in the heatmap. Although some cells were already committed to the neural lineage and situated in the expected medial position, or presented a transitioning transcriptional profile in line with findings on early spatial patterning[34,35], most of the cells co-expressed neural and non-neural markers and had a pluripotent-like undecided stem cell profile despite being spatially located either medially or laterally (Supplementary Fig. 3b-b"). In sum, these results suggest that the expression of pluripotency genes is not abruptly lost after gastrulation, but rather is maintained in the ectoderm until late neurulation stages. Importantly, the spatial distribution of the subgroups was consistent in all analyzed parallel embryos as shown by mapping back the pseudo-colored cells into the respective original images (Supplementary Figs. 2 and 3). Our findings thereby suggest that the progressive loss of the ectodermal pluripotency-like signature proceeds in concordance with a much slower sepration of the ectodermal domains than previously thought, consistent with recent scRNAseq and imaging reports of a gradual ectodermal patterning process that is completed only near the end of neurulation stage[19,21,36,37].

## Pluripotency-like signature is maintained in the ectoderm throughout neurulation

The heatmaps from scMST revealed important information on the expression of the stem cell genes (Fig. 1e and Supplementary Figs. 2 and 3). First, the pluripotency factors did not exclusively cluster together at any developmental time point suggesting their expression patterns are not identical and that each gene may play additional individual roles during early development that does not require co-expression with other pluripotency markers, as previously known from several other contexts[38,39]. Second, consistent with its known role in neural induction at the neural plate after gastrulation[40], the heatmaps show that ectodermal Sox2 expression clusters together with neural progenitor markers and therefore was considered a neural stem cell rather than a pluripotency marker in our study. Thus, we focused on co-expression of *Nanog*, *PouV /Oct4*, and *Klf4* to monitor the transcriptional signature of pluripotent-like cells due to their known collaborative roles in epiblast and embryonic stem cell pluripotency[29,30], which from now on will be referred to as pluripotency-like signature. Based on the stem cell groups we detected by scMST (Fig. 1 and Supplementary Figs. 2, 3), we reasoned that if stemness were to be maintained from blastula stage onwards, continuous co-expression of the pluripotency genes would be readily detected. To test this, we next focused on solely monitoring the expression of the pluripotency genes by identifying cells with scMST z-scores above the mean for all three pluripotency genes simultaneously, which were z-scored separately for each developmental stage (Fig. 2a, b). The results show that at the end

of gastrulation (HH5), the ectoderm is comprised of numerous cells with a pluripotency-like signature and that these cells are maintained broadly throughout the entire ectoderm until the neural fold stage. Spatial restriction of these stem cells to the dorsal neural tube was only detected at the end of neurulation (Fig. 2a and Supplementary Fig. 4a), indicating that the pluripotency-like signature is not restricted to the NC domain during the period between gastrulation and neural fold closure. After identifying the presence of cells with a pluripotency-like signature in the developing ectoderm at each stage individually, we sought to explore the temporal dynamics of the cells by examining the pluripotency gene expression signature across all stages of development. To do so, we pooled the cells from all the four stages and performed a gene-wise z-score normalization to compare expression levels between stages. Using the same criteria as before, we selected cells co-expressing the three pluripotency genes above the mean and plotted their distribution across all stages. As expected, we observed that cells with the pluripotency-like signature are found throughout the ectodermal patterning process and majority of those cells are found in HH5, comprising 44.4 % of the total number of 5415 cells. At the end of neurulation, the embryo has grown significantly, and only 2.2% of the total number of 8253 cells maintain the pluripotency-like signature, which overlapped with cells that present a transcriptional profile of specified neural crest cells that appear only at this latest time point (Fig. 2c, d), and thus recapitulated our results from the analysis of individual stages.

The expansive expression of pluripotency genes throughout the ectoderm led us to speculate that the purpose of these presumably plastic "pan-ectodermal stem cells" might facilitate a gradual patterning of the ectoderm by maintaining competence to form all ectodermal domains. To test this, we next utilized a similar approach to that described above, ie. scMST z-scores were used to select pan-ectodermal stem cells that presented a pluripotency-like signature together with the NNE marker *TFAP2A* and the future CNS marker *Sox2*, the two transcription factors that were also recently shown to co-bind during neural crest development[41] (Fig. 2b and Supplementary Fig. 4b). The results show that ~70% of the cells with the pluripotency-like signature (Fig. 2a) also displayed a pan-ectodermal stem cell gene expression profile and were found in all the ectodermal domains until being restricted to the dorsal neural tube at the end of neurulation (Fig. 2b and Supplementary Fig. 4c). It is worth noting that while the majority of the pan-ectodermal stem cells reside in the NC domain at the end of neurulation, some cells spatially overlapped with the neural stem cells thereby highlighting the plasticity of the DNT (Fig. 1d). Furthermore, maintenance of the pluripotency-like signature appears to be unique to the ectodermal germ layer as no cells were detected in the mesoderm or endoderm (Fig. 2a). Thus, both analysis approaches from the scMST data in Fig. 2a–d show existence of cells throughout

the developing ectoderm germ layer with a pluripotency-like signature that co-express ectodermal domain markers. Additionally, we compared the stem cell populations identified in Fig. 1 (transitioning and undecided groups defined by the heatmap) with those identified in Fig. 2b (defined by selection of genes co-expressed above mean levels) to determine the extent of overlap in cell identities. As expected, >90% of the pan-ectodermal stem cells were found to be the same cells as the transitioning and undecided stem cell groups. (Supplementary Fig. 4d).

**Bulk-RNA-sequencing analysis shows continuous expression of pluripotency-like signature in the developing neural crest cells**
These results prompted us to hypothesize that the potential for the formation of the exceptionally high NC stemness is maintained in the ectoderm past the gastrulation stage by the continued co-expression of pluripotency genes. To test this, we used bulk-RNAseq to analyze dissected cranial NPB/NC regions (red regions in Fig. 1a) from embryos collected only ~1 h apart from each other to ensure detection of subtle transcriptional changes from gastrula to end of neurulation stage. Replicates consisted of pooled tissue from a minimum of five embryos, with three to four replicates obtained for each stage. As expected, the samples were readily distiunghished by developmental time (Fig. 2e), and accordingly, the gene ontology term representations, list of differentially expressed genes and overall change in gene expression reflected the early steps of NC development (Fig. 2f and Supplementary Fig. 4e). In line with the scMST results, a linear graph shows continued expression of stem cell genes including the three pluripotency factors *Nanog*, *PouV/Oct4* and *Klf4* (Fig. 2g), and no gaps were detected between any stage. As a validation of the sample dissection, the expression of the early NPB marker Pax7 started to rise soon after gastrulation, and the expression of the NC specifier gene Sox10 became prominent at the late neural fold stage, as expected (Fig. 2g). Because pluripotency circuitries are well studied in embryonic stem cells, we next asked if other components of the pluripotency complex were expressed in the developing NC. For this, we plotted other genes obtained from an unbiased list of core pluripotency factors and the linear graphs showed continuous expression of eleven additional stem cell factors throughout the neurulation stages, thus further supporting our findings (Fig. 2h). Next we performed a time-course differential expression analysis, which further demonstrates the gradual decrease of the pluripotency genes accompanied with a gradual increase of the neural crest markers with an overlapping intermediate period of the two during the neural fold stages (Fig. 2i and Supplementary Data 1, 2). Additionally, volcano plots show that the pluripotency genes are differentially expressed throughout all stages of premigratory neural crest development as compared to the later, migratory NC stage (Fig. 2j). Similarly, we compared consecutive stages (pooled according to HH stages, Supplementary Fig. 4f) in order to further validate these findings, which show that while Nanog is differentially expressed between the first early ectoderm stages HH5 and HH6, its expression does not significantly change between the neural fold stages HH6, HH7, and HH8, and it is downregulated only at the end of neurulation at HH9, whereas PouV/Oct4 expression does not significantly change from after gastrulation until HH7, and it is downregulated between HH7 and HH8 and then maintained unchanged until the NC emigrates (Supplementary Fig. 2e). Simultaneously, genes reflecting NC induction at the NPB are differentially expressed starting at HH6, and the increase of expression levels continues throughout the ectodermal maturation, and is complemented with later neural crest specification markers at HH8 as well as a downregulation of the premigratory / EMT marker genes like Axud[42] and Draxin[43] at HH9/HH10 (Supplementary Fig. 4f). Similarly, volcano plots between the early ectoderm and late neural folds stages (HH6 vs HH8) shows that while there is no change in the expression of the pluripotency genes, the NC genes are differentially expressed, indicating they are not continuously expressed but

rather activated at the late neural fold stage at the onset of NC specification and are thus not likely to maintain the excpeptionally high NC stem cell potential from gastrula stage as previously suggested[17] (Supplementary Fig. 4f, g). The linear plots revealed a slight overall decrease in transcript numbers of the pluripotency genes during the neurulation process, which raised the question of whether the transcript levels are biologically relevant and translated into protein. Immunofluorescence confirmed that Nanog was readily detectable in the NC domain at the end of neurulation (Supplementary Fig. 4h). Together, scMST and bulk RNAseq analyses show that co-expression of pluripotency genes is not lost in the ectoderm upon gastrulation but is maintained in the entire ectodermal germ layer as it is patterned into its functional domains. The pluripotency-like signature, as evaluated by co-expression of core pluripotency genes, is eventually restricted to the dorsal neural tube (neural crest and neural stem cell domains) at the end of neurulation.

**scRNAseq supports scMST findings on ectodermal pluripotency-like signature**
To further validate our results, we collected scRNAseq samples at the midbrain level from the equivalent developmental stages used for scMST and analyzed their transcriptomes by using the 10x genomics platform (Figs. 1a and 3a). From two independent samples per stage (consisting of pooled tissue from three to six embryos per sample), we captured five to twenty thousand cells per sample for sequencing (Supplementary Fig. 5a, b). As expected, we identified cell clusters reflecting all three germ layers (Fig. 3b, d, Supplementary Fig. 5c, d, Supplementary Data 3). Consistent with the scMST results, feature plots revealed that pluripotency gene expression was largely restricted to the ectoderm (Supplementary Fig. 5c, e, f). For a higher resolution analysis of the expression of pluripotency genes during ectoderm development, we subclustered the ectodermal cells (Fig. 3e). *Nanog*, *PouV/Oct4* and *Klf4* were continuously expressed in the ectodermal cells during neurulation (Fig. 3f). To gain further understanding on the dynamics of the transcriptional process of the pluripotency factors, we took advantage of the RNA velocity algorithm that calculates the ratio between unspliced and spliced transcripts. A predominance of unspliced RNA is a sign of active transcription of the respective gene[44]. We reasoned that if stemness were maintained from the gastrula stage, we would expect to see high velocity for the pluripotency genes throughout the neurulation process. Indeed, pluripotency gene mRNAs were predominantly unspliced and positive velocity was thus found at all four developmental stages. In contrast, active transcription of the *Tfap2A* control gene showed reduced velocity at the end of neurulation as demonstarated by the predominance of spliced mRNAs (Fig. 3g, h). Finally, we asked whether pan-ectodermal stem cells could be identified from the scRNAseq data set, similar to what we discovered with scMST. We used module scoring to visualize cells with a pan-ectodermal stem cell signature featured on the UMAPs of the ectodermal subset. As expected, we detected cells with a high pan-ectoderm score at all four developmental stages (Fig. 3i). Next, we ascertained the locaton of the specified NC cells to learn whether the pan-ectodermal stem cells identified in the scRNAseq data set eventually reside in the NC domain at the end of neurulation as suggested by the scMST results in Fig. 2b. Indeed, cells with a high score for the mature NC module, consisting of genes known to be expressed at the end of neurulation (premigratory, specified NC) stage, were detected at the end of neurulation and showed a similar pattern to the pan-ectodermal stem cells in the UMAP (Fig. 3i). Furthermore, quantification showed that at the end of neurulation stage, 76% of the cells with a high score for the NC signature also had a high score for the pan-ectodermal stem cell signature (Fig. 3i and Supplementary Fig. 5g). scMST revealed some pan-ectodermal stem cells also in the neural stem cell population in the dorsal neural tube (Figs. 2b and 1d). To look for signs of this in the scRNAseq data, we created a module also for

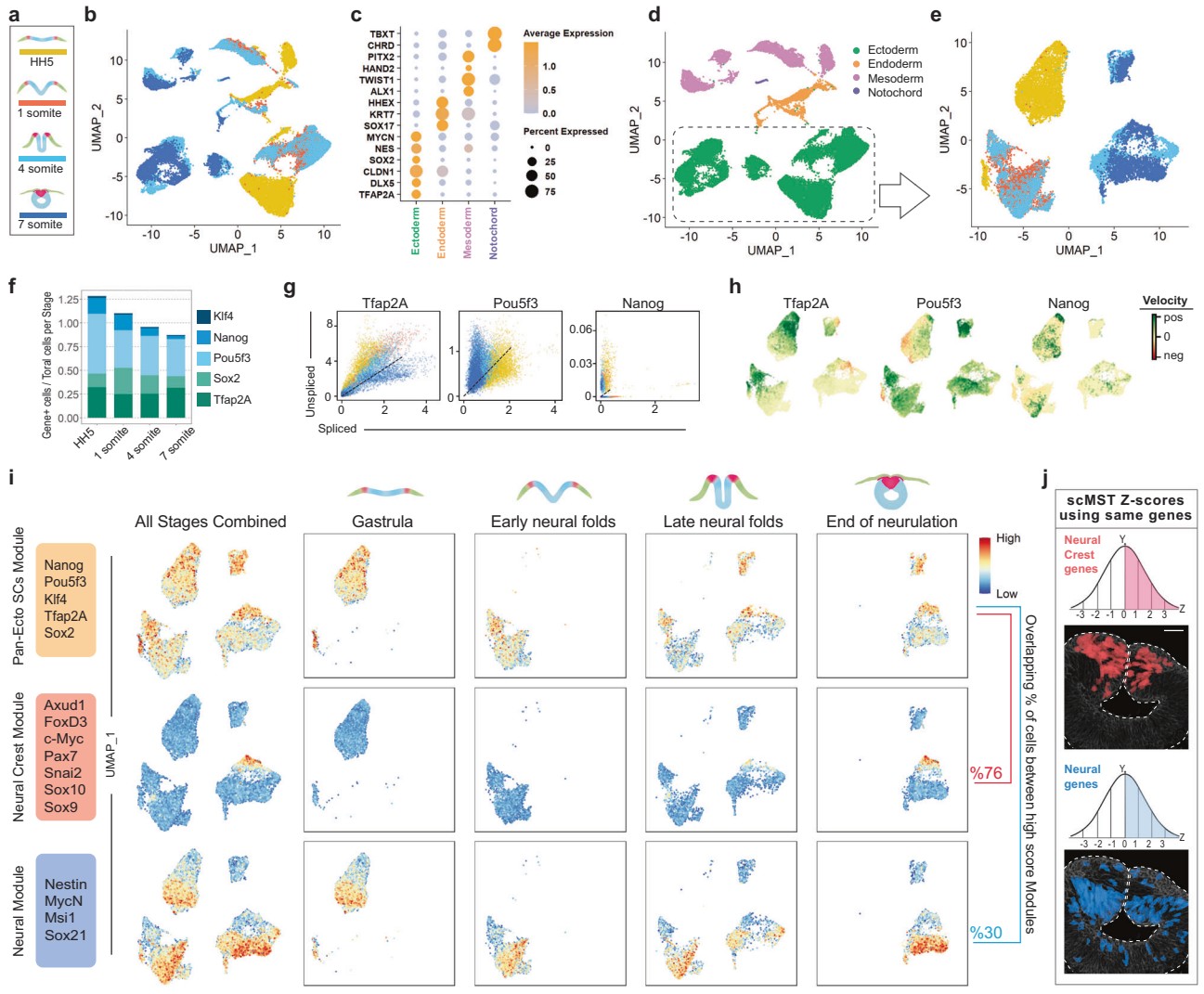

**Fig. 3 | scRNAseq data substantiate maintenance of pluripotency-like signature in the pan-ectodermal stem cells. a** Color-coding according to developmental stage of scRNAseq samples in figures from **b** to **g**. **b** UMAP of all data colored according to developmental stages. $n = 2$ biological replicates for each stage. HH Hamburger Hamilton chicken stage, SS Somite Stage. **c** Based on enriched genes in each cluster shown in the dot plot, cells in the UMAP separate according to germ layer identities. **d** Re-colored UMAP representing each germ layer. **e** Ectoderm cluster was subsetted for a detailed analysis. New UMAP is colored to show each developmental stage in the ectoderm group. **f** Bar plots highlight continued expression of pluripotency genes in the cranial ectoderm from gastrula to end of neurulation. **g** RNA velocity plots show a higher ratio of unspliced RNA for *Nanog* and *PouV/Oct4* throughout developmental stages indicating continuous active transcription, whereas *Tfap2A* transcripts at the end of neurulation are

predominantly spliced. **h** Positive velocity (green) in cells that show higher abundance of unspliced mRNA in UMAP. **i** Modules reflecting Pan-ectodermal stem cells, NC and neural genes with their expression scores shown on the feature plots on combined UMAP and on each stage separately. At the end of neurulation (7ss) stage, 76% of the cells with high NC score, and 30% of the ones with high neural score also share high score for pan-ectodermal stem cells. **j** A complementary figure generated from scMST analysis using the same developmental stage (7ss) as scRNAseq data; cells co-expressing the same set of genes (with expression values above the mean, z-score > 0) used in scRNAseq NC and neural modules, are visualized in order to provide a spatially-resolved perspective of the cells highlighted by module scoring. $n = 4$ biological replicates. Scale bar = 30 µm. Pan-Ecto SC Pan-ectodermal stem cells.

CNS neural cells. Indeed, 30% of the cells with a high neural module score also scored high for pan-ectodermal stem cells (Fig. 3i and Supplementary Fig. 5g). In addition, to complement the scRNAseq data and to provide spatial reference in the embryo, we utilized scMST data to visualize cells that co-express the genes that we used for the respective scRNAseq modules Fig. 3j. Next, we investigated the differeces in the transcriptional profile of the pan-ectodermal stem cells as compared to the rest of the ectodermal cells ("Others") (Fig. 4a). As expected, pan-ectodermal stem cells readily expressed genes representing all three domains. While the NNE and NPB markers were differentially expressed in the pan-ectoderm population (Fig. 4b), neural stem cell markers are equally expressed in both populations (Fig. 4c).

However, the "other" ectodermal cells show statistically significant increase in the expression of committed neural progenitor markers (Fig. 4c, Supplementary Data 4), supporting the idea that NC forms from the pan-ectodermal population, and suggesting that the maturation of the neural lineage is initiated before commitment of the other domains during the ectoderm patterning process, as also shown by scMST (Supplementary Fig. 3b, b' and b"). The scRNAseq data thus corroborate our scMST and bulk RNAseq results, which together indicate that the pluripotency-like signature is maintained throughout the ectoderm during neurulation, which we hypothesize is necessary to inhibit lineage commitment during a gradual ectodermal patterning process. Furthermore, we hypothesize that this is the underlying

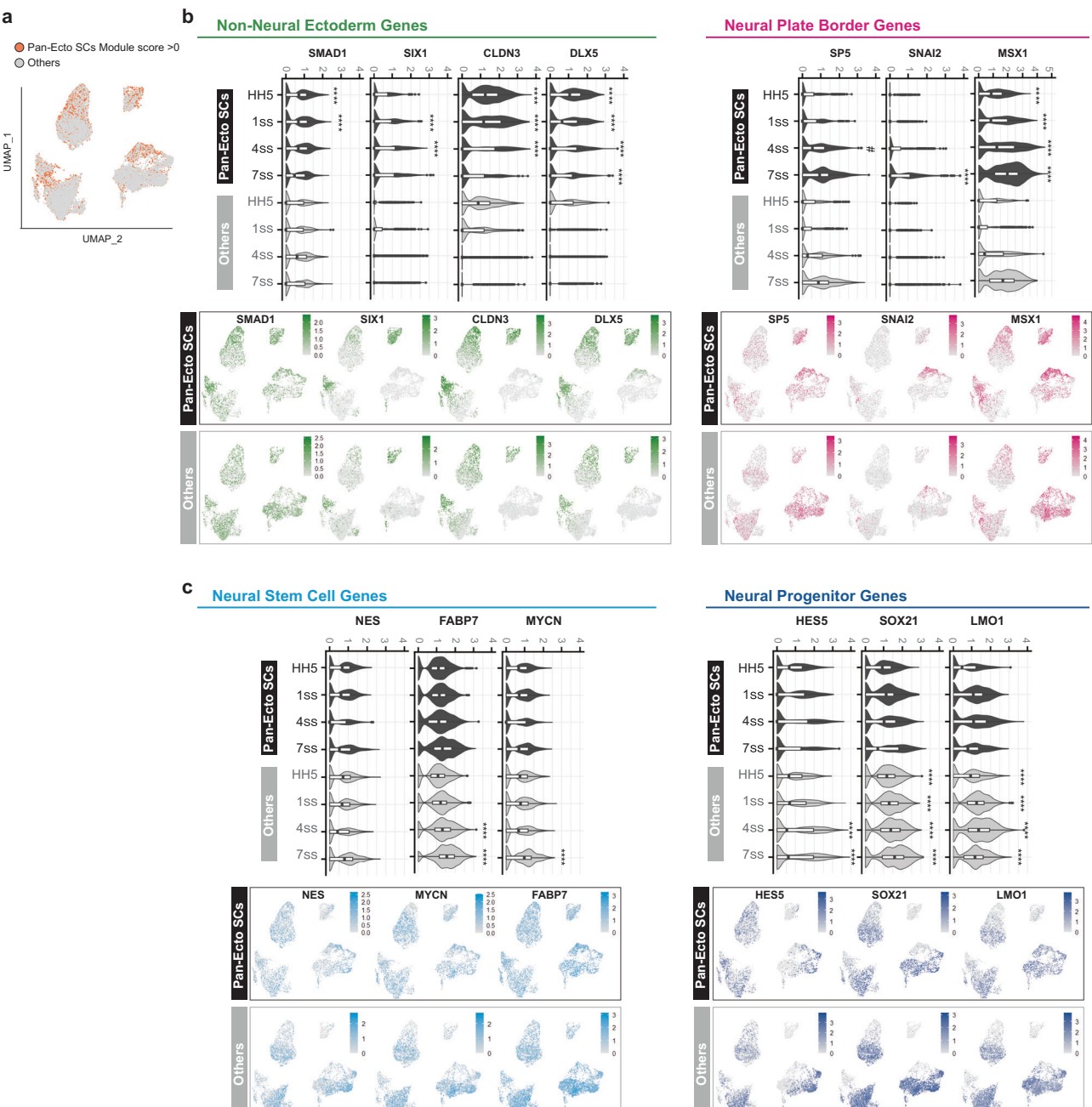

**Fig. 4 | Transcriptional profiles of Pan-ectodermal stem cells vs "other" ecto-dermal cells. a** Cells with scores >0 for the Pan-ectodermal Module (see Fig. 3i) were marked and the rest of the ectodermal cells were labeled as 'others' for comparison analysis. **b** Non-neural ectoderm (future epidermis) and neural plate border (future neural crest) markers are predominantly expressed in the pan-ectodermal subset. Violin plots demonstrate expression levels of differentially expressed individual marker genes within a single developmental stage that represent the non-neural ectoderm and neural plate border from gastrula to end of neurulation stage. Black bar indicates median value (FindAllMarkers function (Wilcoxon rank-sum test), LFC threshold = 0.2, min.pct = 0.25, *p*-adj value < 0.0001 = \*\*\*\* *p*-value < 0.0001 = #). Feature plots demonstrate the expression patterns of the individual marker genes. **c** CNS neural stem cell genes are readily expressed in both groups while more committed CNS neural progenitor markers are pre-dominantly expressed in the "others" population, as shown by using the same statistical critia as described in **b**. Pan-Ecto SC = Pan-ectodermal stem cells. The minima, maxima, centre, and percentile values along with the exact *p*-adj and *p*-values are listed in Supplementary Data 4. *n* = 2 biological scRNA-seq replicates for each developmental stage.

mechanism that enables the potential for the formation of NC that possesses an exceptionally high stem cell capacity in the dorsal neural tube at the end of neurulation.

### Pluripotency genes mutually regulate genes involved in stem cell functions

Our results suggest that co-expression of pluripotency factors Nanog, PouV/Oct4 and Klf4 in pan-ectodermal stem cells maintains stemness and allows plasticity during ectodermal patterning. However, the pluripotency genes may also have independent roles. To investigate this, we individually knocked down the three pluripotency genes using bilateral electroporation of a translation blocking morpholino on one side and a control morpholino on the contralateral side at gastrula stage (HH4) and analyzed the difference in gene expression in the NPB domains at neural fold stage (HH7/HH8·) by bulk RNA sequencing (Fig. 5a). The samples (pool of 6-7 NPB domains per sample)

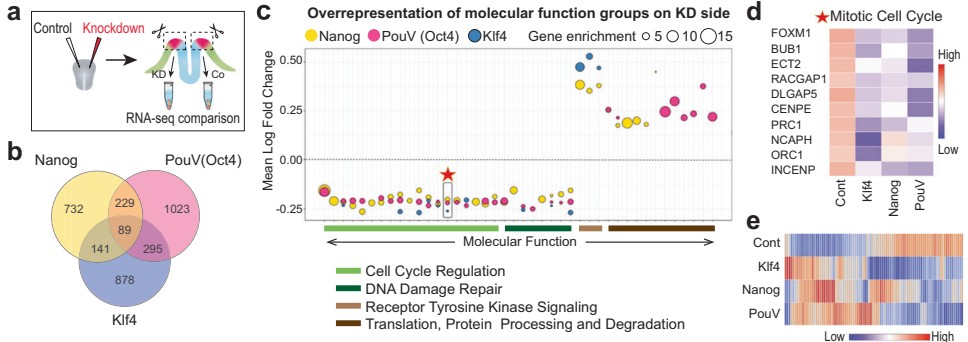

**Fig. 5 | Knockdown of Nanog, Pouv/Oct4 and Klf4 affects stem cell functions in the ectoderm. a** Experimental design for knockdown by using two-sided injection of translation blocking morpholinos. *n* = 4 biological replicates per condition and *n* = 9 biological replicates per control. **b** Venn diagram depicting the number of genes that are differentially expressed due to individual knockdown of pluripotency genes highlighting 89 mutually affected genes as well as genes affected by only one or two of the knockdowns. **c** The common downregulated molecular functions of all three individual knockdowns include genes involved in cell cycle regulation and DNA damage repair. A weighed scatter plot highlights trends seen among the list of molecular functions in overrepresentation tests for each knockdown (KD) group among the 89 shared genes as compared to pooled contralateral control sides. The gene enrichment (bubble size) depicts the number of differentially expressed genes found in a particular category and the number of times greater this pool size is compared to what is expected in a random list of genes. Gene enrichments were obtained using statistical overrepresentation test, and *p*-values were calculated using Fisher's Exact test and adjusted using the Bejamini-Hochberg false discovery rate (FDR) method for multiple test correction. **d** Heatmap showing an example of the genes included in the mitotic cell cycle function marked with a star in **c**. **e** Heatmap highlighting the expression pattern of the 89 genes affected by knockdown of all three pluripotency genes.

segregated according to treatment (Supplementary Fig. 6a). First, to understand the mutual roles shared by the three pluripotency genes, we generated a Venn diagram of all the 3387 differentially expressed genes, which showed that 89 were shared across each of the three knockdowns (Fig. 5b, Supplementary Data 5). The functional enrichment analysis revealed overrepresentation of 27 functional groups related to mitosis, cell cycle and DNA repair, in all of which the expression of genes relevant to the subgroup are downregulated in all three knockdowns, highlighted by an example heatmap of group 14 (marked with a star) consisting of mitotic cell cycle genes (Fig. 5c–d, Supplementary Fig. 6b). Of these functional groups, 30% (8/27) were downregulated following all three pluripotency gene knockdowns, and 78% (21/27) were downregulated following at least two of the knockdowns (Fig. 5c). In contrast, several other cellular functions such as those related to translation, protein processing and degradation were primarily (67%; 10/15) affected by loss of only one of the genes (Fig. 5c and Supplementary Fig. 5b). Furthermore, individual molecular function over-representation plots also revealed general cell functions related to cell proliferation, growth, survival and stemness (Supplementary Fig. 5c–e). Overall, a heatmap depicting all the differentially expressed genes affected by knockdown of all three pluripotency genes supports the conclusion that while the outcomes caused by the individual knockdown of the three pluripotency genes are similar, unique gene-specific responses were also detected (Fig. 5e). Additionally, regarding several cellular functions, different components of the same pathway are targeted by a different pluripotency gene, as shown in an example in Supplementary Fig. 5f. We also performed Pax7 immunostainings on the embryos to investigate the impact of the individual knockdowns of the pluripotency genes on the NPB domain (Supplementary Fig. 7a). Our results show that loss of Nanog and Klf4, respectively, decreased levels of Pax7 expression at the NPB domain as compared to the contralateral control side, whereas decreased levels of Oct4/Pouv translation increased the amount of Pax7 expression in the NPB region. Embryos injected with control morpholino on both side showed no significant difference between the two sides (Supplementary Fig. 7b). Furthermore, downregulation of Nanog reduced the thickss of the developing neural tube, while loss of Klf4 and PouV did not cause a statistically significant difference as compared to the control-injected side (Supplementary Fig. 7c, Supplementary Data 6). Combined, these results suggest that the pluripotency genes function together to regulate cellular processes that enhance stem cell capacity

to ensure a proper ectodermal patterning process and neural crest formation, although further experiments to decipher the complexity behind the respective functional individual vs shared roles of the pluripotency factors during ectodermal patterning and NC development will be necessary to pinpoint the respective molecular mechanisms.

## Pluripotency-like signature is found in trunk NC

These findings prompted us to hypothesize that co-expression of Nanog, PouV/Oct4 and Klf4 is a general requirement for NC development unrelated to the future lineage commitment of the NC cell. This is relevant since NC cells at different axial levels in the embryo are known to give rise to different cell types[8]. To test this, we performed multi-channel fluorescent in situ hybridization (FISH) on older embryos (HH13, a stage during which the posterior neural folds are developmentally equivalent to the head region of a HH7-9 embryo) to address whether the pluripotency genes are expressed in the trunk NC, which does not give rise to mesodermal-like derivatives. Indeed, we detected continuous co-expression of pluripotency genes throughout the posterior ectoderm (Fig. 6a, b) suggesting that their expression does not dictate a specific NC fate. Additionally, cross-sections from the trunk level show that while individual *PouV* /*Oct4* and *Klf4* expression is detected in the entire developing neural tube, co-expression of all three pluripotency genes is restricted to the dorsal neural tube at the premigratory neural crest stage (Fig. 6b), similar to our findings in the head region (Fig. 1d), which also further supports the idea of the pluripotency genes serving both mutual and individual roles in the developing ectoderm.

## Broad ectodermal pluripotency-like signature is found in mouse embryos

Finally, studies on the mechanism of NC stemness formation have been performed in multiple model organisms, predominantly the mouse, frog, and chicken[17,18,22,26]. Since species-specific differences are known, we asked whether evidence for the main discovery of our work, the broad ectodermal maintenance of a pluripotency-like signature, exists in a mammalian model by performing FISH on equivalent developmental stages of mouse embryos. The results on cranial sections of (E8) mouse embryos show co-expression of *Nanog, PouV/Oct4* and *Klf4* throughout the entire ectoderm (Fig. 6c) similar to the chick embryo (Fig. 6d). While our findings provide initial evidence for a

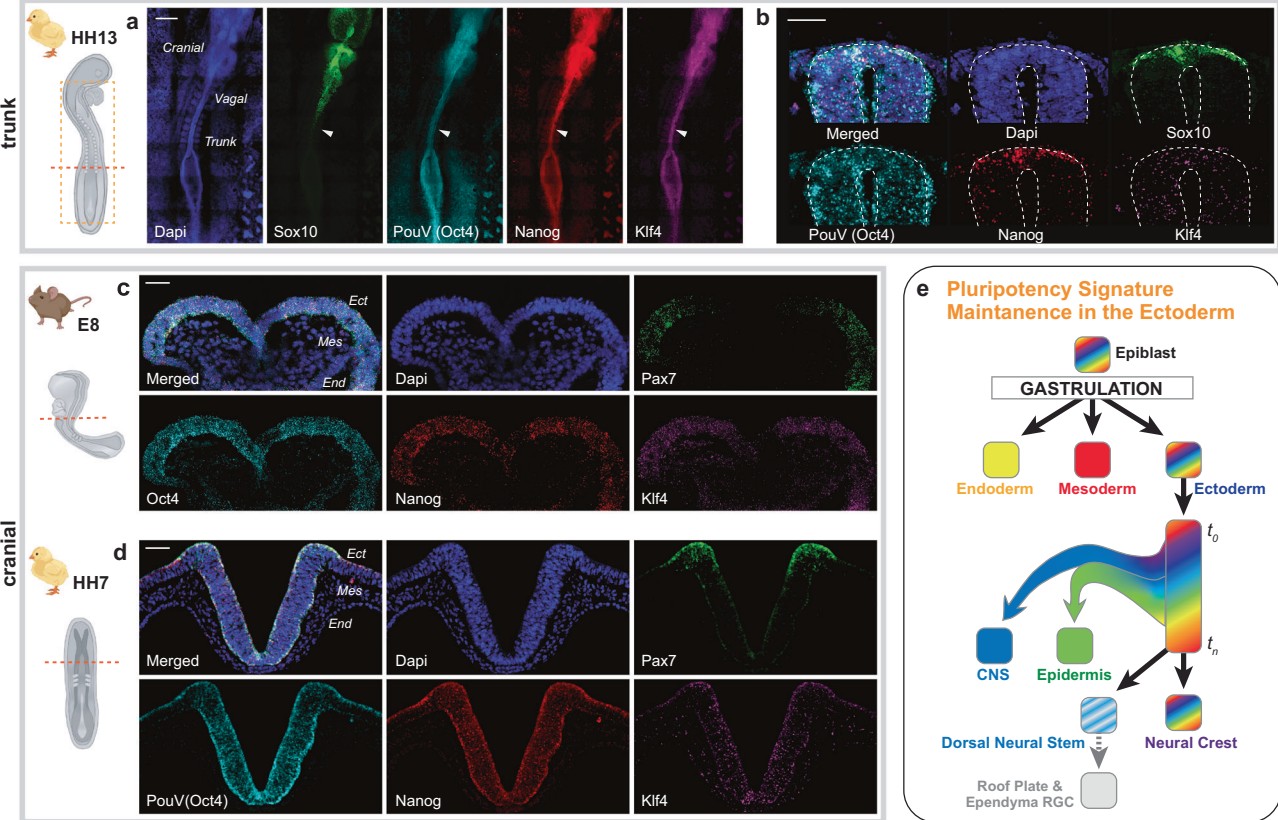

**Fig. 6 | The pluripotency-like signature of the neural crest is detected at multiple axial levels, and broad ectodermal expression of pluripotency genes is found also in mouse. a** Whole mount multicolor fluorescent in situ hybridization shows expression of *Nanog* (red), *PouV/Oct4* (cyan), and *Klf4* (purple) in the trunk neural folds and plate of a HH13 chicken embryo. The arrow points to the end of neurulation level where the neural crest is premigratory as indicated by the onset of *Sox10* (green) expression. *n* = 4 biological replicates. HH Hamburger Hamilton chicken stage. Scale bar = 300 μm. **b** Cross-section from the trunk level marked by arrow in a. Note that while co-expression of the pluripotency genes is restricted to the dorsal neural tube, *PouV/Oct4* and *Klf4* are expressed more broadly and may play additional roles during CNS development. *n* = 4 biological replicates. Scale

bar = 30 μm. **c** Cranial level cross-section from a mouse embryo at E8 shows broad expression of the pluripotency genes in the entire ectoderm. Pax7 demonstrates the neural crest domain. *n* = 4 biological replicates. Scale bar = 50 μm. **d** Cranial cross-section of a HH7 chick embryo shows similar expression pattern of pluripotency genes as seen in mouse. *n* = 4 biological replicates. Scale bar = 50 μm. Chick (yellow) and mouse (brown) cartoons included in the images were created using BioRender.com. **e** A schematic depicting our hypothesis on ectodermal maintenance of pluripotency-like signature after gastrulation until the end of neurulation, highlighting the mechanism that enables the formation of the exceptionally high neural crest stem cell potential.

conserved mechanism across vertebrates, future studies will be necessary to evaluate the extent of similarity in different model organisms including the mouse.

## Discussion

We used single cell level gene expression analysis techniques to investigate both transcriptional and spatiotemporal changes reflecting cell fate commitment at the midbrain level during the ectoderm patterning process post-gastrulation. We find that co-expression of pluripotency genes is not lost during gastrulation nor restricted to the NC domain. On the contrary, while mesoderm and endoderm lose pluripotency, we find cells with a pluripotency-like signature located throughout the ectodermal germ layer. The pluripotency-like signature is co-expressed with both future CNS and epidermal domain markers, leading to their designation of "undecided pan-ectodermal stem cells". We hypothesize that the continued co-expression of pluripotency genes is: (1) a requirement for maintaining cellular plasticity during the gradual patterning of the ectoderm, (2) the mechanism by which the NC domain acquires its potential for an exceptionally high stemness enabling it to form derivatives beyond "ectodermal capacity", and (3) the mechanism by which the dorsal neural tube maintains plasticity as it transitions from NC to roof plate and ultimately the radial glia (neural stem cells) in the ependyma[45]. Our study, using three independent

high-resolution techniques, proposes that stemness is retained from the blastula stage (Fig. 6e) and thus resolves some of the controversy arising from previously published studies on NC stemness formation[17,18]. Our data demonstrating that NC specifier genes are first expressed at the end of neurulation, and that they are not continuously expressed in the ectoderm throughout neurulation, do not support a previous suggestion that the NC gene expression circuitry maintains stemness[17]. Furthermore, our finding that pluripotency genes are continuously expressed throughout this period also does not support the idea that they are reactivated in the NC[18]. Instead, we propose that it is the continuous expression of pluripotency genes that is essential for the formation and maintenance of NC stem cell potential, which does align with the proposed principle idea of stemness maintenance from gastrula stage[17] as well as with the long-standing hypothesis that all ectodermal derivatives are specified after gastrulation, also supported by the previous scRNAseq data[19]. However, our data reveals a different mechanism to explain how the neural crest gains its pluripotency-like signature that is mediated by the pluripotency GRN and expands the stem cell potential to the entire ectoderm rather than being restricted to the neural crest domain. Our finding of broad spatial distribution of the cells with a pluripotency-like signature throughout the ectoderm aligns with recently reported transcriptional and lineage tracing data on a gradual ectodermal patterning

process[19,21,36,37]. Rather than a prompt fate commitment process right after gastrulation, our scMST and RNAseq data show a range of uncommitted and intermediate lineage commitment states as neurulation proceeds, which we hypothesize is enabled by the pluripotency genes maintaining an undifferentiated state that gradually shifts towards commitment.

Importantly, we detect expression of pluripotency genes at multiple axial levels suggesting their necessity for NC formation and ectoderm patterning. Furthermore, knockdown of the respective pluripotency genes affected overlapping gene regulatory circuitries involved in proliferation and stemness associated functions. These findings do not align with recent reports that link pluripotency-like features only to the anterior part of the embryo because of the unique ability of the NC to form the craniofacial skeleton in the vertebrate head[18,22]. However, stem cell potential is not necessarily equivalent to the final selection of the cell types the stem cell ultimately produces. In other words, the full potential may not always be utilized. Although the NC does not contribute to ecto-mesenchymal derivatives in the trunk region of the commonly used animal models including chick, frog, mouse and zebrafish[13,46–50], reports from turtle embryos show trunk NC-derived bone in the plastron[51], and transplant studies of axolotl trunk NC have shown their contribution to the branchial arch cartilage[52]. These reports indicate that the high stem cell potential may be an ancestral feature of the NC, initially maintained at all axial levels but lost secondarily in different vertebrate embryo models, according to species-specific needs (see also review[53] for recent discussion). Furthermore, our findings do not rule out the possibility of additional, individual respective roles for the pluripotency genes later in NC development, such as supporting ecto-mesenchymal lineages at emigrating and migratory stage, as recently reported[18,22]. Importantly, the pluripotency-like signature defined by co-expression of core pluripotency genes is not to be confused with their individual expression, such as Oct4 detected both in the maturing neural ectoderm and mesoderm[23,38,39], and further context dependent functional studies are needed to distinguish the individual roles from the concerted action.

Recent discoveries propose a role of pluripotency genes in participating the transition fom pluripotency to committed lineages by co-binding to lineage factors that redirect their binding to new enhancer sites, leading to regulatory interference from other factors inhibiting them to activate downstream targets, and eventually a gradual repression of the pluripotency gene transcription[41,54–56]. We show a comcomitant gradual loss of the pluripotency-like signature and increase in fate commitment during the ectodermal patterning process, suggesting possible involvement of similar epigenetic changes. On the other hand, neural crest cells do posses an exceptionally high pluripotent-like stem cell potential as judged by their ability to form derivatives beyond the ectodermal capacity. Our multiple lines of investigation combined argue for the original role of the pluripotency genes in enhancing stemness (and inhibiting lineage committement) throughout the ectodermal patterning and neural crest specification process, allowing speculation of a dual role for the pluripotency factors in the ectoderm in both maintaining stemness and assisting gradual fate commitment. Future studies are also needed to investigate whether the neural crest cells truly are pluripotent or something in between pluripotency and multipotency (Pleistopotent[8]). Finally, our current hypotheses are built on transcriptional profiling, and future functional studies are required to test all the different aspects of our proposed model.

Lastly, our finding of sustained, broadly distributed ectodermal co-expression of pluripotency genes long after gastrulation has ceased challenges the dogmatic viewpoints of gastrulation in which pluripotency is lost during the formation of the three germ layers. It is worth noting that the formation of ectoderm is very different from endoderm and mesoderm, which both ingress from the pluripotent epiblast (primitive ectoderm) and lose pluripotency in the migration

process, while less is known about the detailed molecular signatures that induce the commitment of definitive ectoderm to form from the remaining epiblast layer, which continues to be maintained as an epithelial sheet[31]. Perhaps this morphological difference has enabled the entire ectoderm with the unique feature of maintaining pluripotency-like characteristics after gastrulation to ensure a faithful ectodermal patterning process, and leading to formation of the NC in vertebrates.

## Methods

### Chicken
Fertilized chicken eggs were obtained from the University of Connecticut (UCON) poultry farm (CT, USA) and Lassilan Tila (Tuusula, Finland), incubated at 37 °C to reach the desired stage according to Hamburger and Hamilton (HH). See Fig. 1a for the exact stages used for each experiment. All chicken embryos collected for the study are younger than 3 days of age, usage of which does not require ethical approval (The Finnish Act on Animal Experimentation (62/2006), The Institutional Animal Care and Use Committee(IACUC) of NIH).

### Mouse
Wild-type C57BL/6 J mice were obtained from the Jackson Laboratory. For timed matings, noon on the day a copulation plug was detected was considered to be 0.5 days postcoitus. Embryos were collected in a sex-unbiased manner. All mice were maintained under specific pathogen-free conditions at the NIH animal care facility. The mice are housed no more than 5 adult mice to a cage. With Care fresh Bedding from Envigo, placed on a Lab Product Rair rack that houses 160 cages. The racks have supply air and exhaust air, air exchanges per hour is 35. Light cycle for our facility is 14 on 10 off. Rooms are kept at 68–78 F, humidity is between 30% and 70%. Cages were changed weekly. Mice were cared for, and all experiments were approved by the Administrative Panel on Laboratory Care, and the Institutional Animal Care and Use Committee(IACUC) of NIH. Mice were maintained on a normal chow diet.

### Single Cell Multiplex Spatial Transcriptomics (scMST) pipeline
The scMST pipeline was modified from our previously published technique Spatial Genomic Analysis[26,28], which was originally designed to analyze a single field of view at a single developmental time point. To govern analysis of multiple fields of views per sample and at multiple developmental stages, scripts that enable comparison of gene expression across all samples were added. Detailed instructions and codes are available in Github (https://github.com/KerosuoLab/Pajanoja_2023).

In brief, multiplexing is based on serial single molecule fluorescent in situ hybridization (smFISH) rounds on cryosections that are collected on silane coated coverslips (see below). For each gene of interest, up to 24 short DNA probes (minimum 13 probes when designing of more was not possible, Supplementary Data 7) that target the coding sequence of the same mRNA were used. The 3′ end of each probe was linked with a HCR initiator sequence (B1, B2, B3, B4 or B5[57]) via a 4 nt long linker sequence resulting in a 60 nt-long probe. Expression of five genes was captured on each round using a spinning disc microscope with six channels (see below). At the end of each round, signal was removed by using DNase I, followed by another round of hybridization by using probes to five new genes applied on the same sample. All hybridization and imaging steps were done in hybridization chambers (Grace bio labs, 50ul) that were mounted on the coverslips. In order to verify RNA integrity, the initial hybridization probe set was repeated after the multiplex routine, yielding a signal recovery rate exceeding 70% (Supplementary Fig. 1c). Furthermore, the probe sets and the order of hybridization rounds was the same for all experiments. As the last step of the imaging rounds, cell borders were captured by using immunolabelling, which is a critical step for the success of the machine learning algorithm-based 3D cell segmentation and quantitative dot counting of individual RNA transcripts within

individual cells. For the scMST analysis, each cell with quantitative expression data of the thirty genes was pooled into a heatmap that clusters them into unbiased subpopulations according to their transcriptional profile. The cell annotations are based on the heatmaps generated by using hierarchical clustering analysis. The spatial location of the cells within the hierarchically clustered subpopulations of interest was then visualized on the original tissue section images by using pseudo-coloring based back-mapping. This allows observation of individual cells in their respective subpopulations within the original spatial context of the sample. Importantly, our analysis methodology does not involve stitching of the individual fields of views; to avoid potential errors or artifacts that may arise from the stitching process such as losing the position of dots (transcripts), the images are stitched for visualization after the expression analysis is complete.

The following specific changes were made to the original pipeline:[26,28] First, in order to minimize uneven illumination issues, which occur due to artifacts in the imaging process, images were preprocessed using the two-step filtering macro command[58]. In the first step, the macro utilizes Fiji BaSiC plugin, a retrospective image correction method used to correct uneven image illumination[59]. The plugin corrects flat field in each channel by performing low-rank and sparse decompositions. The second step uses Fiji 3D median filter with 3x3x3 pixel kernel to reduce noise and enhance spot detection. Second, we replaced the signal processing step of our previous version with K-means clustering, an unsupervised machine learning algorithm that clusters dots based on their signal intensity, providing a more robust approach (Supplementary Fig. 1a). The purpose of this step is to exclude unspecifically bound probes (displayed as low intensity) as well as image artifacts that may occur in the images (displayed as low or extremely high intensity). The distribution of dots across different intensity levels is visualized via using scatter plot by plotting their mean intensity values on one dimension and their intensity variability on the other dimension. The dots are color-coded based on their assigned k-means clusters, which facilitates the identification of grouping patterns in the data. After clustering the dots, specific clusters are selected to determine the optimal true signal. To determine the optimal number of clusters for k-means clusters, we first used the Elbow plot and MATLAB's "evalclusters" function for each gene. However, these methods do not always consider dataset complexity due to imaging artifacts. Therefore, it was sometimes necessary to manually inspect the data distribution to select the appropriate number of clusters. The main goal is to accurately identify the true signal by distinguishing it from non-specific probe binding and artifacts. This optimization of this step provides a more unbiased selection of the transcripts as compared to the previous version of simply cutting the signal after a certain value as it takes into account the distribution of the signal intensities and identifies the clusters that are most likely to contain the true signal. Third, the MATLAB code was further optimized to accommodate analysis of samples with multiple fields of views. The gene expression of the individual samples was first z-scored across all three fields of views per each embryo, followed by another z-scoring when the scores of all embryos were pooled together, therefore providing a more accurate and consisting level of "high" or "low" expression of a given gene within each cross section (see Source Data). The sample size of our scMST analysis is $n$ = 4 embryos, cossisting of 1-2 cross sections per embryo (Supplementary Fig. 1b). Ultimately, the pooled data is visualized in a heatmap. The heatmap was generated from two-dimensional hierarchical clustering analysis, which employs cosine similarity as the metric between pairwise objects in the distance matrix. Subsequently, based on the resulting unbiased clusters, cell annotations to subpopulations according to their transcriptional profile were generated. Stage by stage analyses (stage specific heatmaps in Fig. 1d–f, Fig. 2a, b and Supplementary Figs. 2 and 3) were done in this manner. The count matrix and cell IDs for each cell cluster were exported in order to create UMAP plots using the "umap" package in Python.

For stage by stage normalized scMST data, selection and pseudo-coloring of cells that have a z-score >0 (expression higher than the mean) was used to determine cells with overlapping pluripotency gene expression (*PouV/Oct4*, *Nanog* and *Klf4* in Fig. 2a) as well as highlighting pan-ectodermal stem cells (*PouV/Oct4*, *Nanog*, *Klf4*, *TfAp2A* and *Sox2* in Fig. 2b). The same approach was used to complement neural crest and neural modules created in scRNAseq data (Fig.3j). Specifically, we used the same set of genes that were used in the modules, FoxD3, c-Myc, Pax7, Snai2, Sox10, Sox9 for neural crest, and Nestin, MycN, Msi1 for neural modules, respectively. The data set was filtered for the respective genes with a z-score >0 and the cell IDs that show overlapping filter results for all respective genes of interest were intersected. The rationale behind this approach (Fig. 3j) was to provide the reader with a spatially-resolved perspective of the gene expression patterns for these particular cell types.

Additionally, to gain insight into the expression levels of each gene across different developmental time points, all samples were pooled and z-scores were calculated across all developmental stages, allowing for comparison of gene expression levels (violin plots in Supplementary Fig. 1d, and for selection and pseudo-coloring of the cells of interest according to co-expression of genes as shown in Fig. 2c, d).

## Coverslip coating for scMST
The following changes were made to the sample preparation as compared to our previously published protocol:[26,28] in the optimized protocol, glass surface preparation and silane coating of the slides was done as described[60]. Briefly, slides were incubated in 50% (v/v) nitric acid for 25 min, followed by 200 nM NaOH for 15 min to make glass surface more conductive for silane binding. Silanization of the slides was performed with 1% (w/v) Triethoxysilylbutraldehyde (TESBA) made in ethyl alcohol for 5 min at room temperature. The slides were rinsed with 100% ethanol twice and once with distilled water, and heat-dried in an oven at 64 °C for 4 h or overnight. Next, the slides were treated with (0.1 mg/ml) Poly-l-lysine (Sigma) in PBS for 20 min at room temperature, followed by three rinses with PBS. The coverslips were then air-dried and kept at 4 °C in an airtight storage for no longer than 4 weeks.

## Chick embryo sample collection
For scMST, samples were collected on Whatman filter papers, fixed in 4% PFA overnight, and washed 3 times with PBS-0.2% triton, dehydrated in ethanol and stored in −80 °C. The embryos we incubated through a sucrose gradient (5% 30 min), 15% 4 h at 4 °C and embedded into OCT as previously described[26,28] and 20 μm thick transverse cryosections were cut from the midbrain level onto silane coated coverslips.

For bulk RNAseq experiments, embryos were collected on Whatman filter papers in PBS and midbrain level neural plate border and specified neural crest cells, respectively, from left and right sides were manually microdissected using tungsten needles, immediately placed into RNA Lysis Buffer (RNAqueous™−4PCR Total RNA Isolation Kit, Thermofisher, Waltham, MA, USA) and stored at −80 °C. Replicates consisted of pooled tissue from at least five to seven embryos. Four biological replicates were used for samples HH5-HH6, 1ss-5ss, 7ss, and 9ss, while samples 6ss, 8ss, and 10ss had three biological replicates each. For HH5 and HH6 replicates, rectangular patches lateral to the primitive streak were collected according to midbrain neural crest fate mapping as described[34]. For premigratory stages beyond HH6, neural plate borders or neural fold apices corresponding to the midbrain neural crest domain were excised. Beyond 7ss, dorsal neural tubes containing neural crest from the midline to migratory front were collected.

The scRNAseq samples were collected by dissecting midbrain slices (covering all germ layers) from the desired stage by using micro

scissors, which were pooled from three to six embryos per sample (two replicates for each stage) and dissociated with a multi tissue dissociation kit (kit 3,130-110-204, Miltenyi Biotec) at 37 °C for 20 min with intermittent pipetting to achieve a single cell suspension.

## Immunostaining of cell membranes
A cocktail of E-cahderin (BD Transduction 610181), B-catenin (Abcam ab6391) and N-cadherin (MNCD2, Developmental Studies Hybridoma Bank, AB_528119) primary antibodies were used to label the cell membranes. Secondary antibodies were AlexaFluor647 for E-cahderin and AlexaFluor488 for B-catenin and N-cadherin. The signal from all three antibodies from the two channels was combined to define the cell borders. Comprehensive details regarding the antibodies utilized can be found in Supplementary Data 8.

## scMST Imaging
Imaging was performed using an Andor Dragonfly 200 spinning disk confocal system coupled to a Zeiss AxioObserver (Zeiss, Thornwood, NY). Coverslips were placed onto sealed coverglass rectangular chamber (ASI Imaging, I-3078-2450) in order to minimize errors for aligning images in each imaging round. A 63X apochromat water immersion objective (NA 1.27) was used for imaging cross sections. An Andor integrated laser engine provided the excitation light using 405 nm (100 mW), 488 nm (150 mW), 561 nm (100 mW), 594 nm (100 mW), 640 nm (140 mW), and 730 nm (30 mW) laser lines, and using suitable emission wavelengths for each fluorophore (DAPI, AF 488, Cy3B, AF 594, Alexa 647, and Cy7), respectively. A Photometrics Prime 95B CMOS (Photometrics, AZ) camera was used in 12-bit mode with 3X gain (13×13 μm pixels). Exposure times and relative laser intensity varied based on sample/fluorophore brightness ( ~ 500 ms and ~40% for each fluorophore). A Z piezo stage (ASI Imaging, Eugene, OR) allowed rapid imaging in Z. Images were collected every 0.5 μm over a 25 μm distance. All components were controlled by Micro-manager version 1.4.22 using the Andor driver and was programed by ADD. DAPI staining was imaged during the first round in addition to the five sets of probes (405 nm). For the final round of immunostaining, Alexa647 (640 nm) for E-cadherin with 500 ms with 20% laser power, and Alexa488 (473 nm) for B-catenin and N-cadherin cocktail was used with 400 ms 20% laser power. Each cross section was captured using 3 fields of frames (boxes with dashed lines in Fig. 1a) with the help of the automated position software built in the microscope. All images were aligned in our custom Matlab script by overlaying the images from each hybridization round and choosing the coordinates for the ideal alignment. The alignment process involves a trial and error approach, as it is necessary to visually inspect the aligned images to ensure proper alignment. Final pseudo-colored visualizations of the embryo sections are stiched together after all analysis is complete.

## Morpholino knockdown and RNA extraction for bulk RNAseq
Translation blocking morpholinos were designed to target the 5′ UTR in close proximity of the ATG for the respective genes: Nanog (CATGGTCGGGACGACACCTCCAG), PouV (CCGAGAGCTGCCTCCATGCTA), and Klf4 (CCGTCGTCCCGCCGAGGAGAGT). A control morpholino was designed to account for non-specific effects from electroporation (CCTCTTACCTCAGTTACAATTTATA). Morpholinos were conjugated on the 3′ end with carboxyfluorescein. Morpholinos were diluted to a 1.0 mM concentration and electroporated together with an empty pCIG vector as carrier DNA (1 μg/μL). Injections were two-sided, targeting the ectoderm in HH stage 4 (gastrula) chicken embryos, with control morpholino on the contralateral side. Chicken embryos were collected on Whatman filter papers and electroporated using 5.3 V and 5 pulses (50 mA/100 mA), as previously described[61]. Embryos were then incubated on individual petri dishes in thin albumin until they reached HH7 (1–3 somites).

For the bulk RNAseq analysis, embryos were then dissected to isolate the dorsal neural folds, which were immediately placed into lysis buffer. RNA was extracted from the tissue using the RNAqueous-4PCR Total RNA Isolation Kit (AM1914) according to kit directions. Extracted RNA was then ethanol precipitated and resuspended in nuclease free water before storing at −80 °C.

## Pax7 immunostaining and neural tube thickness image analysis
For the Pax7 immunostaining (Hybridoma Bank Pax7 AB_528428, 1:10), the embryos were fixed in 4% PFA at 4 °C overnight, permeabilized with PBS-Triton 0.2% for 1 h at room temperature, and immunostained as whole mount embryos, embedded in 8% gelatin and cryo-sectioned and imaged as previously described[61]. The intensity of Pax7 expression was quantified by measuring Integrated Density (IntDen) in Fiji. The experimental morpholino side was compared to the contralateral control morpholino side (an average value of at least five sections per embryo = n), and the ratio was compared to the ratio of control embryos injected with a control morpholino on both sides. To assess the effect of the morpholino knockdowns on neural tube thickness, the width of each section was measured at the dorsal, middle, and posterior parts of the neural tube (as shown in Supplementary Fig. 7c), using the same number of n (an average value of at least five sections per embryo). These measurements from the three points were then averaged together. Subsequently, the width of the experimental morpholino side was compared to that of the contralateral control morpholino side, and the resulting ratio was compared to the ratio obtained from control embryos injected with a control morpholino on both sides. Statistical analysis was performed using an unpaired Student's $t$-test.

## RNA sequencing analysis
**Bulk RNAseq for 12 developmental stages of the neural crest domain.** Bulk RNA-seq was performed by the NIDCD/NIDCR Genomics and Computational Biology Core (GCBC). Libraries were created using the Nextera XT method. Neural Plate border / Neural crest domain samples were collected from 12 different developmental stages from end of gastrulation to end of neurulation (Fig. 1a), each timepoint consisted of three to four replicate samples, and each replicate consisted of pooled samples from five to seven embryos. All samples were run simultaneously on an Illumina NextSeq 2000 configured for 55 PE reads. The counts were generated by mapping via STAR aligner (v2.5.2a). Reads are 55x55bp in length, and samples contain between 43.7 and 87.5 million uniquely mapped reads to gene regions. Counts were filtered to remove all genes which did not have more than 5 reads in at least 1 of the samples. After performing PCA analysis, the samples were observed to be clustered together with their corresponding replicates and separated from other time points, indicating no batch effects. The downstream analysis was subsequently performed (Fig. 2e). Differential expression analysis was carried out using DESeq2[62]. Samples were pooled as shown in Supplementary Fig. 4d, then The DAVID[63,64] resource tool was used to find Gene Onthology (GO) terms using upregulated genes (LFC > 0.75). Fold enrichments were obtained using statistical overrepresentation test, and $p$-values were calculated using Fisher's Exact test and adjusted using the Bejamini-Hochberg false discovery rate (FDR) method for multiple test correction. To perform a comprehensive time course analysis, DESeq2 LRT (Likelihood Ratio Test) was employed and the earliest developmental stage, HH5, was set as the reference level using the "relevel" function. The resulting top 250 up- and downregulated genes were plotted on a heatmap with $P$-adj value < 0.05, LFC threshold 0.75 and variance stabilizing transformation (VST) was calculated for line plots with shaded error ribbons (Fig. 2g–i). Volcano plots were generated for differential gene expression analysis using the Wald test comparing pooled stages in Fig. 2j, and Supplementary Fig. 4f, g ($P$-adj value < 0.05, LFC threshold 0.75).

### Bulk RNAseq for morpholino knockdown of pluripotency genes

Gene counts were produced by the NIDCD/NIDCR Genomics and Computational Biology Core using the STAR aligner in standard mode. The reads were mapped against a current ENCODE chicken release and the mapping parameters were derived from the GENCODE project. Each sample type consisted of four replicates of the respective experimental side samples, and two to four replicates of the control side samples (total of 9 biological replicates) (Supplementary Fig. 6a) where each replicate was pooled from neural plate border regions of six to seven embryos. Counts were filtered to remove all genes that did not have more than 5 reads in at least 1 sample. Then counts were normalized by library size and analyzed with principal component analysis (PCA). A PCA1/PCA2 graph revealed an apparent batch effect preventing meaningful interpretation of the read counts, which was removed by using the RUVseq (remove unwanted variation)[65] function as follows: Remove Unwanted Variation with replicate Samples (RUVs) was used to remove the batch effects by using factor analysis on the control samples. Additionally, one thousand in-silico control genes, identified as those with the lowest coefficient of variance among all samples, were passed to the function. Transformation of the data using RUVs revealed a PCA1/2 plot showing clear gene expression signatures for each KD group and control group. After normalization, differential gene expression (DGE) analysis was performed using DeSeq2, where each KD group was compared to all control samples. This revealed many genes, which showed a statistically significant change ($p$-adj < 0.05) in expression levels between the KD groups and control, most of which were unique to only one KD group. Up- and downregulated genes for each morpholino list were used for the overrepresentation test via PANTHER (GO biological process)[66]. Fold enrichments were obtained using statistical overrepresentation test, and $p$-values were calculated using Fisher's Exact test and adjusted using the Bejamini-Hochberg false discovery rate (FDR) method for multiple test correction.

### 10X Chromium scRNA-seq

Single cell sequencing was performed at FIMM Single-Cell Analytics unit supported by HiLIFE and Biocenter Finland. Single cell gene expression profiles were studied using 10x Genomics Chromium Single Cell 3'RNAseq platform. The Chromium Single Cell 3'RNAseq run and library preparations were done using the Chromium Next GEM Single Cell 3' Gene Expression version 3.1 Dual Index chemistry.

The sample libraries were sequenced on Illumina NovaSeq 6000 system using read lengths: 28 bp (Read 1), 10 bp (i7 Index), 10 bp (i5 Index) and 90 bp (Read 2). All 8 samples were sequenced together in a single sequencing run to avoid any potential batch effects. Fastq files were generated by using the 10x Cell Ranger (v7.0.1) count pipeline of Cell Ranger as well as by performing alignment, filtering and UMI counting. Finally, Fastq files were mapped to chicken genome using a custom GRCg6a Gallus gallus genome annotation file that was constructed by extending selected 3'UTRs in order to account for incorrect annotations.

Data cleaning, normalization and scaling was performed as follows: downstream analysis was carried out using Seurat package (v3.0.1)[67] in R. SoupX[68] and DoubletFinder[69] packages were used to filter out ambient RNA and doublets, respectively. Cleaned samples were merged and named based on their original file names. Following the standard Seurat workflow, artifact cells were further excluded by removing any cells that expressed more than 4000 genes and high mitochondrial content (>0.4%). Gene expression matrices were normalized and scaled using the NormalizeData and ScaleData functions. During the scaling, Cell Cycle Difference scores, which was calculated by difference of S from G2/M-phase specific gene scores via 'CellCycleScoring' function, were regressed out. Then, principal component analysis linear dimension reduction was conducted using using the 'ElbowPlot' function (resolution 0.2, dims 1:15). Clustered cells were visualized via UMAP plot and manually annotated based on differentially expressed genes in each cluster (FindAllMarkers function, logfc.threshold = 0.2, min.pct = 0.25). The clusters that were identified as "Ectoderm" were extracted and re-clustered using the 'subset' function and given colors based on their individual stages. Bar plots were used to demonstrate expression levels of selected genes. Each bar represents cell counts that express the gene of interest individually, and the values were normalized by total cell counts in each developmental stage or germ layer, respectively (Fig. 3f and Supplementary Fig. 5f). Module scores were calculated based on individual lists of selected genes, respectively, using the AddModuleScore function in Seurat. For in depth analysis, cells in the ectoderm UMAP were labeled based on module score as "Pan-ectoderm" (Pan-ecto score > 0) vs "Others" (Pan-ecto score < 0). Differentially expressed genes between Pan-ectoderm vs Others were calculated via stage by stage comparison (FindAllMarkers function (default setting: Wilcoxon rank-sum test), logfc.threshold = 0.2, min.pct = 0.25). Ratios of unspliced vs spliced RNA were calculated via scVelo dynamical modeling pipeline[44] in Python. For each sample, Loom files containing spliced and unspliced RNA expression matrices were created using the Velocyto.py command line tool. Ectoderm subsets of HH5, 1 somite, 4 somite and 7 somite stages were selected for the analysis.

### Antibody production

Custom Nanog and FoxD3 Monoclonal antibodies (Genscript Antibody Services, Piscataway, NJ) were commercially generated against full-length recombinant chicken proteins. Forty parental lines were received and tested by immunoblot and immunohistochemistry using chicken lysates and fixed embryos (FoxD3: HH9-10; Nanog: HH5-6 primordial germ cells). Select parental lines were clonally expanded and monoclonal hybridoma isotypes were confirmed (Pierce™ Rapid Antibody Isotyping Kit, Thermofisher).

### Whole mount fluorescent in situ hybridization amplified by using Hybridization Chain Reaction (HCR)

Chicken (HH6- HH13) and mouse embryos (E7.75–E8.5) were fixed in 4% paraformaldehyde, washed with PBS/0.2% Tween (PBST), dehydrated in methanol, and stored at −20 °C. All experiments involving animals were approved by the NIDCR Institutional Animal Care and Use Committee. HCR split-initiator probe sets were purchased from Molecular Instruments (www.molecularinstruments.com) for chicken Nanog (NM_001146142.1), PouV (NM_001309372.1), Klf4 (XM_004949369.3), Pax7 (NM_205357.1), and Sox10 (NM_204792.1). In situ HCR v3.0 with split-initiator probes was done according to whole-mount chicken protocol[70] with hybridization overnight at 37 °C and amplification overnight at room temperature. Prior to hybridization, later stage embryos (HH12-HH14) were permeabilized with 10 μg/mL proteinase K for 2.5 min. Following amplification, embryos were washed with 5xSSCT, stained with DAPI, and post-fixed with 4% PFA for 15 min at room temperature. Embryos were mounted using ProLong Gold Antifade Mountant and imaged using an Andor Dragonfly spinning disk confocal microscope. For sections, embryos were embedded in gelatin and sectioned at 20 μm.

### Reporting summary

Further information on research design is available in the Nature Portfolio Reporting Summary linked to this article.

## Data availability

All data required to assess the study's conclusions are present in the main text and supplementary materials, and the source data are provided with this paper. Bulk RNAseq (time course) GSE221125, MO bulk RNAseq GSE232432 and scRNAseq GSE221188 datasets have been deposited to in the NCBI GEO database. Source data are provided with this paper.

## Code availability

MATLAB scripts for scMST pipeline and R scripts for all scRNAseq analysis have been deposited into GitHub (https://github.com/KerosuoLab/Pajanoja_2023; https://doi.org/10.5281/zenodo.8234085).

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

## Acknowledgements

We thank Sally Moody and Terry Yamaguchi for helpful comments on the manuscript. We thank Selen Özkan and Heli Takko for their valuable input on the optimization of the coding steps. We thank FIMM Single-Cell Analytics unit supported by HiLIFE and Biocenter Finland for the scRNA sequencing services. We thank NIDCD/NIDCR Genomics and Computational Biology Core for the bulk RNA sequencing services and the Veterinary Resource Core for hosting the mice. **Funding:** This research was supported by the Intramural Research Program of the NIH, NIDCR, NIH ZIA DE000748 (to LK), and The Academy of Finland and Sigrid Juselius Foundation (to LK), and Väre Foundation and Emil Aaltonen Foundation (to CP). This study was also supported in part by the Genomics and Computational Biology Core: ZIC DC000086, the NIDCR Imaging Core: ZIC DE000750-01, and the NIDCR Veterinary Resources Core: ZIG DE000740.

## Author contributions

LK conceived the study. LK and CP designed the experiments. CP, AS, LK, JH, SA, and ZZ prepared the embryo samples. CP optimized the scMST code with help from AD. CP and ADD optimized scMST sample preparation and imaging. CP performed scMST. CP and LK analyzed scMST, scRNAseq and WT bulk RNAseq data with help from DM. JH, RY, SA, and ZZ performed HCR and immunostaining. AS prepared customized antibodies. BO, CP, RY and LK analyzed the KD bulk RNAseq and immunostaining data. LK and CP wrote the manuscript with input from all authors. LK supervised the study.

## Funding

## Competing interests

The authors declare no competing interests.
