## [Peer Review File · Nature Communications]

REVIEWER COMMENTS

Reviewer #1 (Remarks to the Author):

By tracking the transcriptional profile of the ectoderm tissue of the avian embryo, findings of this study pointed to the sustenance of cellular pluripotency in the neural crest (NC) cell progenitors at all axial level till late in the neurulation, and that allocation of the neuroectoderm and the ectomesenchyme is accompanied by time- and space-related transition from the stemness cell state.

A domain of “transient NC stem cell niche” was defined by the expression of three pluripotency factors, Nanog, Oct4 and Klf4. This niche property was found to persist in the ectoderm past the stage of gastrulation and over time became regionalised to a subset of NC cells which are reputed to the NC stem cells. This observation is consistent with the notion that cellular pluripotency is retained by the forerunners of the neural crest cells instead of being re-installed at the allocation of the NC lineage among the neural plate border cells.

Specific points:

Clarify how the presence of transitional stem cell types that express more than one specific lineage markers may be inferred to possess “higher plasticity” or “broader stemness” (page 4). The apparent plasticity (and expression of markers of neighboring domains) might not equate to lack of cell fate restriction to the “committed groups”. The fate of the specific type of transitional stem cells in the three domains: CNS, NNE and NPB has yet to be interrogated in the context of plasticity or “undecided” / “broad” / “high” stemness.

Functional attributes of the pluripotency factors (page 9): While the analysis of functional enrichment may point to the different roles of the three pluripotency factors in the ectoderm, it awaits experimental validation to delineate the relative contribution of the common downstream molecular activity and the “unique gene-specific response” to the regulation of cellular functions that collectively sustains high stemness of the transitioning stem cells/ pan-ectoderm cell” (and for lineage specification) at different stage of neurulation and in the progenitors of various ectodermal derivatives: CNS, NNE NPB.

The gene expression data of the cranial tissue of the E8 mouse embryo may provide support of the broad maintenance of pluripotency in the neural plate/folds. Whether a similar developmental profile of spatial restriction of the “transitional stemness” state is present in the mouse during neurulation remains to be verified.

Minor points:

Abstract: Which are examples of the “endoderm-like cells” that are derived from the neural crest cells?

Clarify if the “plastic” DNTs were constituents of the NC stem cells or the “committed” descendants of the neural stem cell left behind after the emigration of the NCs (page 3-4).

“subdomains of the ectoderm at late neural fold stage have much higher plasticity” (P4 paragraph 1). Does the plasticity reflect cellular heterogeneity in this context?

Clarify the difference, if any, between the “transitioning stem cells”, the four stem cell groups (Fig 1f) and the “pan-ectodermal stem cells” (or “pan-ectodermal cells”), the latter was reputed to “reside” in various restricted domains during neurulation (Page 6). The terminology for these stem cells may have to be harmonised.

Methodology

This study employed several profiling techniques, multiplex HCR, single-cell and bulk RNA sequencing, to generate the transcriptome datasets for mapping the gene expression features to cells at defined spatial position in the embryo by the multiplex spatial transcriptomics pipeline (scMST; Fig. 1b, Methods page 23).

Specific points for consideration / clarification:

1. scMST was performed to obtain the spatial information for a selection of 30 cell markers (Figure 1c), whereas scRNA-seq was performed to capture the full transcriptomes of single cells, and bulk RNA-seq was performed to capture the more differentiation time points as in scMST and scRNA-seq data. The experimental design of omics data generation is appropriate. However, it is not clear how data analysis has properly leveraged these resources as the data generated from each technique appeared were analysed independently and not in an integrated approach.

2. The designation of cell types based on the expression and thresholding of selected makers may be not aligned with the purpose of profiling the transcriptomes of cells at single-cell resolution and population

level by scRNA-seq and bulk RNA-seq respectively. A more data driven approach may leverage the data more productively. A suggestion would be to use scRNA-seq data to impute the transcriptomes of cells profiled in scMST and perform data-driven clustering of cells to identify cell types/cell states.

3. To demonstrate the changes of the expression of marker genes, it may be more informative to show log₂ fold changes with reference to the starting time point (HH5) (Figure 2e).

4. It is not clear if there are any biological replicates in scMST and scRNA-seq data. Batch effect in these datasets should be assessed and not glossed over. The batch effect would have to be mitigated besides simply running through the Seurat pipeline. Biological replicates at one or more time points will be useful for such investigation.

5. Clarify how the k-means clustering was used for achieving unbiased signal selection, and how the number of k=8 was determined.

Reviewer #2 (Remarks to the Author):

This manuscript presents a clear and compelling set of data systematically analyzing, at the level of individual cells in developing chick embryos, the spatial and temporal dynamics of the expression of genes known to play important roles in regulating pluripotency in epiblast stem cells (Nanog, Oct4, Klf4). They focus their analysis on early phases of ectoderm development from post-gastrula stages to the end of neurulation by applying single cell spatial transcriptomics (scMST), in combination with bulk and scRNAseq, to characterize the levels and patterns of expression of these genes.

This work is interesting and important because it is known that pluripotency genes are involved in regulating the formation of neural crest cells (ncc), which represent a migratory population of multipotent cells derived from ectoderm. A fundamental question in the field is how does the expression of these genes, associated with the transcriptional signature of pluripotency, arise in the progenitor population of cells that will form ncc? Based on analyses in different vertebrates several different hypotheses have been put forward: 1) cells are reprogrammed to reactive expression of the pluripotency signature in ncc progenitors or 2) expression of the signature is maintained in relevant progenitor populations during early development. The rigorous analyses in this study provide clear evidence that resolves this issue. Aspects of the transcriptional signature for pluripotency seen in epiblast cells of the early embryo is maintained in the entire ectoderm as it forms and then is progressively restricted to ncc progenitors during neurulation. Using whole mount FISH in situ approaches, they also provide some evidence that the same is true for mouse embryos.

This work is well done, provides compelling experimental support for the spatial and temporal persistence of transcriptional signatures of pluripotency in ectoderm of chicken embryos and highlights the importance of functionally testing whether and how components of this transcriptional signature underlie regulation of the multipotent properties of nccs in different vertebrates.

In principle, I feel this work makes an important contribution to the field. However, it has several significant flaws that need to be addressed before it could be considered worthy of publication.

1) The paper does not accurately reflect what is currently known in the field. In many places they refer to the unexpected persistence of expression of genes in the pluripotency signature beyond gastrulation. In fact this is not unexpected. For example, Osorno et al *Development* 139 (2012) and Lee et al *J. Anatomy* 217 (2010) analyzed expression of Nanog, Oct4 and other genes in the pluripotency signature in developing mouse embryos and show that the levels decline but persist in neuroectoderm into early somites stages of neurulation. The Briggs et al *Science* (2018) reference 4 cited in the paper also suggests that the pluripotency signature is not maintained only in ncc progenitors but in other early neuroectoderm cells. Hence, published data does strongly suggest a model for persistence of a pluripotency signature but it was not systematically examined in detail to clearly define the temporal and spatial dynamics of these patterns. A more balanced presentation of existing data and state of knowledge rather than incorrectly stressing aspects of this work are unexpected or novel needs to be done.

2) The expression data are clear and I have no concerns about the experimental results. However, throughout the text the authors equate transcriptional signatures, gene expression, with functional properties, i.e. pluripotency. They have only analyzed gene expression and there are no functional studies to score for levels of pluripotency.

- For example on page 5 a section title is : “Pluripotency is not lost at the end of gastrulation but is maintained throughout the ectoderm”. They really mean that aspects of the transcriptional signature for pluripotency is maintained. They are really speculating that this translates to the potential of the cells.
- Another example: on page 6: “the ectoderm is comprised of numerous cells expressing a pluripotent signature and that these.... indicating that stemness is not restricted to the NC domain during the period between gastrulation and neural fold closure.” The expression does not indicate stemness is present it is suggestive.
- Another example, page 7: “Together, scMST and bulk RNAseq analyses show that pluripotency is not lost in the ectoderm upon gastrulation but is maintained in the entire ectodermal germ layer as it is patterned into its functional domains.” These data do not show pluripotency is not lost, they show transcriptional correlates of pluripotency are maintained. The authors assume or speculate transcriptional signatures of pluripotency are equivalent to pluripotency itself.

There are many more examples. This kind of imprecise language really detracts from the work. The authors do a nice job in the discussion of speculating in a balanced way what these signatures mean. However, they really should be more precise and accurate in stating that they are really only looking at transcriptional signatures and not equate them with function until the discussion. There needs to be a thorough revision of the text to rectify this issue.

Minor suggestion.

The level of referencing for the first paragraph of the introduction is inadequate. For those not expert in ncc biology there is insufficient background citation and there are a number of outstanding reviews on ncc that could be used to provide more context.

Reviewer #3 (Remarks to the Author):

Through a series of experiments including spatial transcriptomics and functional perturbations, Pajanoja et al set out to test the hypothesis that a pluripotency network is retained in cells in the epiblast that subsequently adopt a NCC identity. They use spatial transcriptomics to demonstrate that factors that operate in the control of pluripotency in embryonic stem cells (Nanog, Oct4 and Klf4), are also expressed broadly in the ectoderm and subsequently retained in neural crest progenitor cell types. This raises the question of what role these factors may be playing in NCCs. They begin to test the role of individual factors through knockdown approaches. Their findings reveal that knockdown promotes changes in gene expression, as observed in bulk RNAseq assays, with computational predictions suggesting these factors play largely similar, yet not identical roles, in these cell types.

Overall, the main claims in the text are not supported by the results presented. The main hypothesis of the paper is not directly tested – that maintenance of a pluripotency network explains the broad potential of NCCs. Transcription factors are known to play context specific roles, and thus, their continued expression through multiple stages of development cannot be used to extrapolate function (c.f. the changing roles of the core pluripotency factor Oct4 in Tiana et al., 2022 PMID: 35857513). From the current data, it is difficult to resolve when the pluripotency network is lost on the basis of expression patterns, without direct functional testing, which the authors have not made clear in the text. Perturbation experiments are attempted to knockdown individual factors, and the mRNAseq results suggest changes in gene expression have occurred, but the consequences are not yet investigated beyond computational predictions. Are cells able to retain the ability to generate different NCC derivatives? If not, what do these cells become? Is this a consequence of pluripotency network changes, or through other mechanisms? Do other cells adopt an NCC identity, in response to establishment of a core pluripotency network? Testing the ability of NCCs to differentiate into different derivatives, in the presence or absence of the core pluripotency network, and definition of what precisely this network is, may help to generate data in support of these hypotheses.

Reviewer #4 (Remarks to the Author):

In this manuscript, Pajanoja and colleagues present a single-cell, spatiotemporal profile of the neural crest (NC). The authors identify that stemness signatures are preserved in the ectoderm and not

necessarily restricted to the NC. The manuscript is well written and the breadth of technologies for this study is impressive. In particular, the authors perform multiplexed single cell spatial transcriptomics (scMST), an iterative approach that enables detection of five genes at a time for repeated rounds of hybridisation up to 30 genes. I have some comments regarding the scMST and scRNA-seq components of the study that should be addressed.

- Did the authors assess for any evidence of technical-driven correlations among genes that were selected to be profiled in the same hybridisation round? If so how was this accounted for in downstream analysis?

- Similarly for individual fields of view, how did the authors ensure that normalisation/thresholding of intensities per colour was adequate, especially for confounding of varying expression patterns spatial regions with technical fields of view? Figure 1d (right) appears to display some vertical lines which might be driven by field of view effects.

- Spatial back-mapping is a visualisation technique where false colour images are created with cells coloured by the cluster to which they have been assigned according to the 30-genes signature. As alluded to in the manuscript, for cells that may be borderline between clusters (e.g. Neural crest stem and Epidermis stem), it would be worth assigning a confidence value to the cluster identity. This could be done by post-hoc supervised learning to extract an associated confidence measure, or alternatively by repeated clustering with a slightly different approach (e.g. gaussian mixture models rather than k-means clustering).

- The authors perform scRNA-seq as a validation of the results obtained from the scMST data. It would be worthwhile directly assessing the concordance between the estimated gene expression profile of scMST and that of scRNA-seq. Additionally, were the authors able to directly integrate these two data together into a joint space? Doing so may enable prediction of additional subclusters or smooth patterns, although this may be limited by the small number of genes profiled and/or the need for z-score normalisation of the scMST data.

- I suggest the authors make the processed scMST data available to readers, either as supplementary or via figshare/zenodo. While the raw images might be large and unwieldy, the cell boundaries, centroids and normalised gene expression values should be quite amenable for public data sharing.

REVIEWER COMMENTS

Reviewer #1 (Remarks to the Author):

By tracking the transcriptional profile of the ectoderm tissue of the avian embryo, findings of this study pointed to the sustenance of cellular pluripotency in the neural crest (NC) cell progenitors at all axial level till late in the neurulation, and that allocation of the neuroectoderm and the ectomesenchyme is accompanied by time- and space-related transition from the stemness cell state.

A domain of “transient NC stem cell niche” was defined by the expression of three pluripotency factors, Nanog, Oct4 and Klf4. This niche property was found to persist in the ectoderm past the stage of gastrulation and over time became regionalized to a subset of NC cells which are reputed to the NC stem cells. This observation is consistent with the notion that cellular pluripotency is retained by the forerunners of the neural crest cells instead of being re-installed at the allocation of the NC lineage among the neural plate border cells.

Specific points:

Clarify how the presence of transitional stem cell types that express more than one specific lineage markers may be inferred to possess “higher plasticity” or “broader stemness” (page 4). The apparent plasticity (and expression of markers of neighboring domains) might not equate to lack of cell fate restriction to the “committed groups”. The fate of the specific type of transitional stem cells in the three domains: CNS, NNE and NPB has yet to be interrogated in the context of plasticity or “undecided” ! “broad”! “high” stemness.

We thank the reviewer for binging up this point, which we have now clarified in the text in the results section as well as in the discussion. Based on our transcriptional analysis, the expression of the pluripotency genes associates with expression of genes that reflect multiple ectodermal domains (NNE/NC/CNS). Cells that only express genes of a single domain do not express the pluripotency genes, which we interpret as a sign of commitment to the respective lineage. On the other hand, cells that co-express the pluripotency genes and also evenly express genes of all domains, we interpret as the cells not having committed to one lineage yet, but they instead still maintain all options available and thus have open chromatin for multiple lineage fates. The transitioning stem cell groups are transcriptionally in between these two states, where one of the ectodermal domain lineages is already dominantly expressed, but the cells have not fully committed yet, since they still co-express lower levels of the neighboring domain markers as well as the pluripotency genes. We hypothesize that the expression of the pluripotency genes is necessary for preventing differentiation, which is essential for allowing the ectodermal patterning process to proceed in a much more gradual fashion than previously anticipated, and to maintain the undecided status in the neural crest cells until the end of neurulation.

Functional attributes of the pluripotency factors (page 9): While the analysis of functional enrichment may point to the different roles of the three pluripotency factors in the ectoderm, it awaits experimental

validation to delineate the relative contribution of the common downstream molecular activity and the “unique gene-specific response” to the regulation of cellular functions that collectively sustains high stemness of the transitioning stem cells/ pan-ectoderm cell” (and for lineage specification) at different stage of neurulation and in the progenitors of various ectodermal derivatives: CNS, NNE NPB.

We agree with the reviewer that analysis of functional enrichment can only provide putative insights to the roles of the respective genes but certainly awaits experimental validation before conclusions on functional mechanisms can be drawn. Accordingly, we have added further comments on the requirement of functional validation to the text (after the functional enrichment results and into the discussion). Understanding the respective roles the pluripotency genes play together and individually during ectoderm development is a natural follow up to our findings in this paper, but we hope the reviewer agrees that these data will consist of years of work and thus an entity of its own and are thus beyond the scope of this paper. To provide some further insight into the separate roles we have added results from morpholino-based knockdowns of the pluripotency genes individually, and we show that loss of Nanog and Klf4, respectively, results in a decreased NPB domain as analyzed by immunostaining of Pax7, whereas loss of PouV/Oct4 results in an expansion of the NPB (new Supplementary Figure 7).

The gene expression data of the cranial tissue of the E8 mouse embryo may provide support of the broad maintenance of pluripotency in the neural plate/folds. Whether a similar developmental profile of spatial restriction of the “transitional stemness” state is present in the mouse during neurulation remains to be verified.

We thank the reviewer for bringing up this point. To gain insight into whether our main finding, the co-expression of pluripotency genes in the entire ectoderm (and not just in the neural crest domain), applies to a mammalian species, we obtained supporting evidence by performing multi-channel FISH to the cranial mouse ectoderm, which indeed showed a similar expression pattern as found in the chick. We agree with the reviewer that to claim perfect similarity of events between the two species, an in-depth transcriptional stage by stage analysis of mouse ectoderm development, similar to our extensive amount of data shown in this paper, will be necessary. However, since the novelty of our work is the discovery of the developmental phenomena of a broad ectodermal pluripotency signature for which we used the chick ectoderm as our model, and since we did find similarities in mouse to support our hypothesis on our finding being a conserved mechanism across at least these two species, we feel examining the details of mouse ectoderm would not bring anything critically relevant in this regard to this current work.

Minor points:

Abstract: Which are examples of the “endoderm-like cells” that are derived from the neural crest cells?

The neural crest is shown to give rise to multiple endocrine type secretory cells (*Stem Cells*, 2021; PMID: PMID: 33017496) including glomus cells in the endocrine gland carotid body (*Histochemie*, 1973; PMID: 4693636, *J Embryol Exp Morphol*, 1975; PMID: 1185098, *Cell*, 2007; PMID: 17956736)), that produce the enzyme Carbonic anhydrase required for the carotid chemosensory responses to CO₂ and to O₂; Chromaffin cells of the adrenal medulla (*Dev Biol*, 1974; PMID: 4140118)) that secrete the stress response hormones adrenaline, noradrenalin and endorphins; approximately 3.3% of hormone producing cells (growth hormone, adrenocorticotrophic hormone, thyroid stimulating hormone, luteinizing hormone) in the anterior pituitary (*Dev Biol*, 1987; PMID: 3817289, *BMC Dev Biol*, 2016; PMID: 27184910, *J Anat*, 2017; PMID: 28026856); and the Calcitonin secreting cells in the ultimobranchial body (*Histochemistry*, 1974; PMID: 4604566, *J Histochem*

Cytochem, 2007; PMID: 17595340, *Development*, 2015; PMID: 26395490). We have now clarified this in the introduction.

Clarify if the “plastic” DNTs were constituents of the NC stem cells or the “committed” descendants of the neural stem cell left behind after the emigration of the NCs (page 3-4).

We apologize for not describing this better and have now clarified this point in the text: The NC has not emigrated yet in our latest time point (when we detect the ore lateral dorsal neural stem cell domain), but rather is about to emigrate. This is why we speculate that the DNT neural stem cell population will soon assist in the rebuilding of the DNT after the neural crest has left. Based on our transcriptional data, we hypothesize that both the neural crest stem cells and the DNT neural stem cells originate from the same pan-ectodermal stem cells that exist earlier in the developing ectoderm before diverging into two distinct stem cell populations in the DNT. Our current data don't suggest the model of the neural stem cells being constituents of the NC stem cells *per se*, but due to the nature of analyzing fixed time points we can't completely rule out that possibility either.

“subdomains of the ectoderm at late neural fold stage have much higher plasticity” (P4 paragraph 1). Does the plasticity reflect cellular heterogeneity in this context?

The higher plasticity in the ectodermal subdomains in this context refers to the transcriptional profile of the stem cells (all the cells that co-express the pluripotency genes, and also, based on their transcriptional profile consisting of genes reflecting multiple ectodermal domains) that are not yet committed to a single ectodermal domain at past mid-neurulation stage. This finding challenges the current model of ectodermal patterning, which suggests that right after gastrulation the ectoderm is fully patterned and thus divided to committed domains (NNE/NPB/NP) (*Cell Mol Life Sci*, 2012; PMID: 22547091), as the systematic existence of these pan-ectodermal stem cells with a pluripotency signature was not known before. Regarding cellular heterogeneity, we don't think the differences between the hierarchically clustered subpopulations are just a reflection of fluctuating cellular heterogeneity but rather representation of cells in different stages of unidirectional cell fate commitment that does not get reversed. This is supported by the fact that the amount of the stem cells is gradually reduced by time both in the number of the different domains they cover, as well as in the proportion they represent of all cells from after gastrulation to end of neurulation (revised Figs. 2a and 2c). In case the reviewer is asking about cellular heterogeneity within each subpopulation, there is always some heterogeneity in the groups regarding levels of expression of individual genes per cells, which we anticipate are constantly fluctuating within in the cells, and that changes of this magnitude are more subtle and fit within the variation of a subcluster.

Clarify the difference, if any, between the ‘transitioning stem cells’, the four stem cell groups (Fig 1f) and the “pan-ectodermal stem cells” (or “pan-ectodermal cells”), the latter was reputed to “reside” in various restricted domains during neurulation (Page 6). The terminology for these stem cells may have to be harmonised.

We thank the reviewer for this comment and realize we need to better clarify the different groups. The different and potentially confusing nomenclature in figure 1 as compared to figures 2 and 3 is chosen because of the way the cell populations are created. In Figure1, by performing stage by stage analysis, the cells of each stage are pooled in a heatmap according to expression of all the 30 genes. This clustering results in distinction of committed cells, transitioning stem cells and undecided stem cells. The difference between transitioning and undecided stem cells is that in the undecided group, genes that reflect the ectodermal

domains are evenly expressed (with some heterogeneity in the subcluster) whereas the transitioning stem cell profiles show a stronger expression of one of the respective domains (NC, epidermis or CNS) while they still express the pluripotency genes and lower levels of genes of the neighboring domains. Fig 1 thus demonstrates the unidirectional ectodermal patterning process from undecided to committed cells. To clarify the scMST results from the stage by stage unbiased clustering, we have now added UMAPs to complement the heatmaps, which nicely reflect the gradual commitment of the cell types during the course of development (new figures added to Supplementary Fig. 2a, 2b', 3a and 3b'). Additionally, we now show the proportions of the stem cells with a pluripotency signature across all the samples in the scMST data in new Fig. 2c). After we discovered that there is continuation of pluripotency gene expression throughout the ectoderm, we took a different approach to focus on the details of the stem cell characteristics. Therefore, the pan-ectodermal stem cells were selected based on co-expression of the pluripotency genes together with the non-neural TfAp2a and neural plate marker Sox2, all selected by expression above mean levels, and the nuances between high and medium expression levels is lost. According to the reviewer's point, we have now calculated the amount of overlapping cell identities between the stem cells defined by these two different approaches (heatmap subclusters in Fig. 1 vs selection of genes above the mean levels in Fig. 2b). The results show that, indeed, the vast majority (~90%) of the pan-ectodermal stem cells are the same cells as the undecided and transitioning stem cells, see new Supplementary Fig. 4d. Due to different transcriptional profiles of how the cell populations were defined in the two approaches, we feel it would not be appropriate to combine the nomenclatures.

Methodology

This study employed several profiling techniques, multiplex HCR, single-cell and bulk RNA sequencing, to generate the transcriptome datasets for mapping the gene expression features to cells at defined spatial position in the embryo by the multiplex spatial transcriptomics pipeline (scMST; Fig. 1b, Methods page 23).

Specific points for consideration / clarification:

1. scMST was performed to obtain the spatial information for a selection of 30 cell markers (Figure 1c), whereas scRNA-seq was performed to capture the full transcriptomes of single cells, and bulk RNA-seq was performed to capture the more differentiation time points as in scMST and scRNA-seq data. The experimental design of omics data generation is appropriate. However, it is not clear how data analysis has properly leveraged these resources as the data generated from each technique appeared were analysed independently and not in an integrated approach.

We thank the reviewer for correctly pointing out that we have not integrated the data from the individual data sets into one metadata entity, which would computationally be extremely challenging and beyond our skills, as there are currently no standard analysis packages available designed for this purpose. The rationale for conducting individual analysis of the datasets was to employ a complementary and validating approach to provide strong evidence of the presence of the pluripotency gene expression throughout the neurulation process. This strategy allowed us to leverage the strengths of each analytical method and create a comprehensive understanding of the temporal transcriptional events occurring in the ectoderm. The advantage of scMST, in addition to providing spatial context, is its ability to capture gene expression from lowly expressed genes, which may be missed from bulk RNA sequencing results due to dilution effects of the bulk sample, and from the scRNA-seq due to low capture efficiency. Furthermore, the current scRNAseq

analysis algorithms are not ideal for grouping stem cells, which expresses low levels of genes of multiple lineages (*Nat Commun*, 2023; PMID: 36878908), and can easily be thrown into multiple more committed clusters according to small heterogeneity within the stem cell population. The strength of the scRNA, on the other hand, is the ability to elaborate the examination of the transcriptional profiles of the cells we identified by using the modules, which we utilized in figure 4, as well as providing details of the active expression with pseudo-time. Finally, the advantage of the bulk RNAseq analysis approach, which we collected samples for from twelve stages just one hour apart from each other in developmental time, was to ensure that we don't miss any, even subtle transcriptional changes during neural crest development, which was essential for testing the two published hypothesis (*Science*, 2015; PMID: 25931449 and *Science*, 2021; PMID: 33542111) as well as for providing important information about the other pluripotency genes that are expressed in the neural crest. We have added a more detailed description of our data analysis method in the revised manuscript to provide more clarity on our approach.

2. The designation of cell types based on the expression and thresholding of selected makers may be not aligned with the purpose of profiling the transcriptomes of cells at single-cell resolution and population level by scRNA-seq and bulk RNA-seq respectively. A more data driven approach may leverage the data more productively. A suggestion would be to use scRNA-seq data to impute the transcriptomes of cells profiled in scMST and perform data-driven clustering of cells to identify cell types/cell states.

We thank the reviewer for this comment and wish to better clarify why we chose to perform the analysis in the current way. Due to the limited number of genes used in the spatial transcriptomics data, it is not feasible to directly compare the cell populations between the two datasets. Importantly, the clustering parameters are very different when the subsetting is based on 30 genes as compared to all the 20K genes from the scRNAseq. In other words, even if we selected cells in R based on the expression of the 30 MST genes, they will still cluster according to the entire transcriptome including differences in metabolic, proliferative, adhesive etc. status, and are likely to produce a different result. On a side note, we are currently performing a deeper analysis on the scRNAseq data to identify the various cell types and cell states that are present in the embryo at these stages. However, the enormous amount of data produced from the analysis is impossible to fit into this project (Pajanoja et al, manuscript in preparation), and we also feel it is beyond the scope of this paper, which focuses in describing the continuous expression of the pluripotency genes and highlights the discovery of pan-ectodermal stem cells.

Instead, we employed a different approach to identify similar cell populations in scRNAseq, based on the expression of key genes that defined those subpopulations in scMST. For this we have created Modules for NC, NPB, Neun and NNE in scRNAseq data and labeled the cells with high scores (module score >0.5) for each module to mimic the subpopulations. Then, we calculated differentially expressed genes between these subpopulations vs rest of the tissue individually. We believe that this approach provides a valuable tool for discovering what other genes those subpopulations may express and thus efficiently utilizes the power of scRNAseq.

Finally, we would like to correct a potential misunderstanding and clarify that z-scoring based thresholding of selected markers was not used for any of the RNAseq methods. We have only used it for the scMST dataset. The main objective of applying z-scoring is to transform the distribution of the gene expression data to a normal distribution with a mean of 0 and a standard deviation of 1. This normalization allows for better comparison of gene expression levels across various samples, since the data is placed on the same scale. Furthermore, z-scoring allows for identification of genes that are expressed at high levels relative to other genes within the same sample. By setting a threshold such as z-score >0, it is possible to identify cells or regions of the sample that have high expression levels (higher than the mean) of a particular gene or gene combination. In Fig3, which shows the pan-ecto cells by using module scoring, we have only used z-scoring

in Fig 3j to complement our scRNAseq findings in order to give a spatial context to the readers. In Fig 3j we selected cells in the scMST data that co-express above mean levels of the same genes as used in creating the respective scRNAseq modules to visualize which cells are selected in the modules, which was entirely separate from the scRNAseq analysis.

3. To demonstrate the changes of the expression of marker genes, it may be more informative to show log2 fold changes with reference to the starting time point (HH5) (Figure 2e).

We thank the reviewer for this point and have now added new data that demonstrate changes in the expression of relevant marker genes by time. We created a big heatmap showing DE genes in reference to HH5 and highlighted neural crest and pluripotency genes (250 up and 250 down regulated DE genes shown in new figure 2i, and we also provide the full DE gene list according to $\text{padj} > 0.05$, $\text{LFC} = 0.75$ (Supplementary Table 5 and 6). Additionally, we generated volcano plots that demonstrate statistically significant differences between all the respective adjacent developmental stages (Supplementary Fig. 4f).

4. It is not clear if there are any biological replicates in scMST and scRNA-seq data. Batch effect in these datasets should be assessed and not glossed over. The batch effect would have to be mitigated besides simply running through the Seurat pipeline. Biological replicates at one or more time points will be useful for such investigation.

We thank the reviewer for checking on this important point and realize we have not written the information clear enough. The samples contributing to the scMST are listed in the original manuscript in Supplementary Figure 1, but we have now also added the sample information into the methods section and figure legends. All scMST data consist of samples from four individual embryos per stage. From each embryo, one or two sections were included (at least five sections per stage in total). The data is pooled and analyzed as a single heatmap, and the data presented in the main figure 1 is spatially recapitulated in all embryos as the respective cells of each embryo are pseudo-colored and the images are shown in supplementary figures 2 and 3 (supp fig 2 in the original manuscript).

All of our RNAseq and bulkRNAseq samples were sequenced together in a single sequencing run to avoid any potential batch effects. Bulk RNAseq samples consist originally of 4 biological replicates (where in each replicate one sample consists of cells from 6-7 embryos), and PCA plots were used to verify that they grouped together, indicating that there is no significant batch effect. The scRNAseq data consists of two biological samples per stage (each sample consisted of cells from 4-7 embryos per sample), and the parallel samples didn't present sample specific subclusters, but instead all clusters were represented evenly in both as expected. We have updated the manuscript to make this information more clear.

5. Clarify how the k-means clustering was used for achieving unbiased signal selection, and how the number of k=8 was determined.

We thank the reviewer for this comment and realize we did not explain the usage of K-means clustering in a sufficiently clear manner. K-Means clustering was utilized in the scMST pipeline to eliminate unspecific probe binding and artifacts from the true signal in our signal processing step as demonstrated in Supplementary Fig. 1a. During the evaluation of our code changes, we tested the optimal number of clusters for k-means clusters using both the Elbow plot and MATLAB's built-in function "evalclusters" for each gene. While these automated methods can provide an estimate of the optimal number of clusters, they do not always take into account the complexity of the dataset, such as imaging artifacts causing intensities to be more scattered. In such cases, we found it necessary to visually inspect the data distribution and manually select the appropriate

number of clusters. When such complexities were introduced, it required dividing the data into more clusters than the automated method suggested. For this step, our primary objective is to accurately identify the true signal by distinguishing it from non-specific probe binding and artifacts. By applying k-means clustering, we can robustly cluster similar data points together and therefore effectively select all data points within the selected range and also remove any clusters with outlier signals, as shown in the new reference image (Supplementary Fig. 1a). We have now added a thorough explanation of this step in the methods section.

Reviewer #2 (Remarks to the Author):

This manuscript presents a clear and compelling set of data systematically analyzing, at the level of individual cells in developing chick embryos, the spatial and temporal dynamics of the expression of genes known to play important roles in regulating pluripotency in epiblast stem cells (Nanog, Oct4, Klf4). They focus their analysis on early phases of ectoderm development from post-gastrula stages to the end of neurulation by applying single cell spatial transcriptomics (scMST), in combination with bulk and scRNAseq, to characterize the levels and patterns of expression of these genes.

This work is interesting and important because it is known that pluripotency genes are involved in regulating the formation of neural crest cells (ncc), which represent a migratory population of multipotent cells derived from ectoderm. A fundamental question in the field is how does the expression of these genes, associated with the transcriptional signature of pluripotency, arise in the progenitor population of cells that will form ncc? Based on analyses in different vertebrates several different hypotheses have been put forward: 1) cells are reprogrammed to reactive expression of the pluripotency signature in ncc progenitors or 2) expression of the signature is maintained in relevant progenitor populations during early development. The rigorous analyses in this study provide clear evidence that resolves this issue. Aspects of the transcriptional signature for pluripotency seen in epiblast cells of the early embryo is maintained in the entire ectoderm as it forms and then is progressively restricted to ncc progenitors during neurulation. Using whole mount FISH in situ approaches, they also provide some evidence that the same is true for mouse embryos.

This work is well done, provides compelling experimental support for the spatial and temporal persistence of transcriptional signatures of pluripotency in ectoderm of chicken embryos and highlights the importance of functionally testing whether and how components of this transcriptional signature underlie regulation of the multipotent properties of nccs in different vertebrates.

In principle, I feel this work makes an important contribution to the field. However, it has several significant flaws that need to be addressed before it could be considered worthy of publication.

1) The paper does not accurately reflect what is currently known in the field. In many places they refer to the unexpected persistence of expression of genes in the pluripotency signature beyond gastrulation. In fact this is not unexpected. For example, Osorno et al Development 139 (2012) and Lee et al J. Anatomy 217 (2010) analyzed expression of Nanog, Oct4 and other genes in the pluripotency signature in developing mouse embryos and show that the levels decline but persist in neuroectoderm into early somites stages of neurulation. The Briggs et al Science (2018) reference 4 cited in the paper also suggests that the pluripotency signature is not maintained only in ncc progenitors but in other early neuroectoderm cells. Hence, published data does strongly suggest a model for persistence of a pluripotency signature but it was not systematically examined in detail to clearly define the temporal and spatial dynamics of these patterns. A more balanced presentation of existing data and state of knowledge rather than incorrectly stressing aspects of this work are unexpected or novel needs to be done.

We thank the reviewer for their supportive views on our work. We appreciate this comment and have now included the suggested and other additional references to the introduction to provide a more accurate view on the existing literature and how that compares to the data in our study.

2) *The expression data are clear and I have no concerns about the experimental results. However, throughout the text the authors equate transcriptional signatures, gene expression, with functional properties, i.e. pluripotency. They have only analyzed gene expression and there are no functional studies to score for levels of pluripotency.*

- *For example on page 5 a section title is : “Pluripotency is not lost at the end of gastrulation but is maintained throughout the ectoderm”. They really mean that aspects of **the transcriptional signature for pluripotency** is maintained. They are really speculating that this translates to the potential of the cells.*

- *Another example: on page 6: “the ectoderm is comprised of numerous cells expressing a pluripotent signature and that these.... indicating that stemness is not restricted to the NC domain during the period between gastrulation and neural fold closure.” The expression does not indicate stemness is present it is suggestive.*

- *Another example, page 7: “Together, scMST and bulk RNAseq analyses show that pluripotency is not lost in the ectoderm upon gastrulation but is maintained in the entire ectodermal germ layer as it is patterned into its functional domains.” These data do not show pluripotency is not lost, they show transcriptional correlates of pluripotency are maintained. The authors assume or speculate transcriptional signatures of pluripotency are equivalent to pluripotency itself.*

There are many more examples. This kind of imprecise language really detracts from the work. The authors do a nice job in the discussion of speculating in a balanced way what these signatures mean. However, they really should be more precise and accurate in stating that they are really only looking at transcriptional signatures and not equate them with function until the discussion. There needs to be a thorough revision of the text to rectify this issue.

We thank the reviewer for bringing up this important point and agree that our terminology was not accurate. We have performed a thorough analysis on transcriptional profiles and states, and have not studied the function of these states. We have now carefully revised the text to clean this discrepancy, and only talk about the potential for pluripotency based on the transcriptional profiling.

Minor suggestion.

The level of referencing for the first paragraph of the introduction is inadequate. For those not expert in ncc biology there is insufficient background citation and there are a number of outstanding reviews on ncc that could be used to provide more context.

We thank the reviewer for this suggestion and have now revised the first paragraph accordingly.

Reviewer #3 (Remarks to the Author):

Through a series of experiments including spatial transcriptomics and functional perturbations, Pajanoja et al set out to test the hypothesis that a pluripotency network is retained in cells in the epiblast that subsequently adopt a NCC identity. They use spatial transcriptomics to demonstrate that factors that operate in the control of pluripotency in embryonic stem cells (Nanog, Oct4 and Klf4), are also expressed broadly in the ectoderm and subsequently retained in neural crest progenitor cell types. This raises the question of what role these factors may be playing in NCCs. They begin to test the role of individual factors through knockdown approaches. Their findings reveal that knockdown promotes changes in gene

expression, as observed in bulk RNAseq assays, with computational predictions suggesting these factors play largely similar, yet not identical roles, in these cell types.

Overall, the main claims in the text are not supported by the results presented. The main hypothesis of the paper is not directly tested – that maintenance of a pluripotency network explains the broad potential of NCCs. Transcription factors are known to play context specific roles, and thus, their continued expression through multiple stages of development cannot be used to extrapolate function (c.f. the changing roles of the core pluripotency factor Oct4 in Tiana et al., 2022 PMID: 35857513). From the current data, it is difficult to resolve when the pluripotency network is lost on the basis of expression patterns, without direct functional testing, which the authors have not made clear in the text. Perturbation experiments are attempted to knockdown individual factors, and the mRNAseq results suggest changes in gene expression have occurred, but the consequences are not yet investigated beyond computational predictions. Are cells able to retain the ability to generate different NCC derivatives? If not, what do these cells become? Is this a consequence of pluripotency network changes, or through other mechanisms? Do other cells adopt an NCC identity, in response to establishment of a core pluripotency network? Testing the ability of NCCs to differentiate into different derivatives, in the presence or absence of the core pluripotency network, and definition of what precisely this network is, may help to generate data in support of these hypotheses.

We thank the reviewer for their comments and wish to clarify a potential misunderstanding. The focus of our work was to thoroughly examine transcriptional changes in the ectoderm during neurulation with the specific question of how does the neural crest gain the previously described transcriptional state in which pluripotency genes are co-expressed at the end of neurulation (*Nat Commun*, 2017; PMID: 29184067, *Methods Mol Biol*, 2019; PMID: 30194538, *Sci Adv*, 2020; PMID: 32494672, *Science*, 2021; PMID: 33542111). To examine this, we performed a full series of complex single-cell level analysis and find that the co-expression of the pluripotency genes within individual cells is continued in the entire ectoderm, and not just the neural crest domain, from gastrula stage onwards, allowing us to hypothesize that the mechanisms by which the neural crest is able to gain the pluripotency gene expression profile is that it is maintained from pre-gastrula stages. We then take further advantage of the transcriptional data sets to test our hypothesis that the cells with the pluripotency signature are undecided between the ectodermal domains, and thus play a role in the ectodermal patterning process, which based on our data seems to be a gradual slow transition rather than a rapid commitment process immediately after gastrulation as previously has been shown (*Cell Mol Life Sci*, 2012; PMID: 22547091). Indeed, we see correlation between the pluripotency signature and co-expression of genes of all three ectodermal domains, whereas the cells with an expression profile of a single domain do not co-express the pluripotency genes. In sum, our results suggest that co-expression of genes of multiple fates at low levels is a sign of stem cell potential and an undifferentiated status. These data suggest that the ectodermal pluripotency is an essential part of the ectodermal patterning, which further suggests that the pluripotency genes are necessary at all axial levels, which is not in line with recent reports linking the pluripotency signature with the cranial-specific ecto-mesenchymal fates (*Sci Adv*, 2020; PMID: 32494672 and *Science*, 2021; PMID: 33542111). To test this hypothesis, we individually knocked down the pluripotency genes and analyzed whether we find changes in the gene expression that would alter the balance of the future neural crest derivatives (gene expression of mesenchymal vs neural vs melanocytic lineages), but we did not find any signs pointing to that direction. Rather, we see that several stem cell associated functions such as proliferation, cell cycle regulation and DNA repair are mutually downregulated as analyzed by using ontology terms, which further supports the idea that the co-expression of pluripotency genes in the developing ectoderm is not related to neural crest lineages per se, but to keep the cells in an undifferentiated state. With this, we want to clarify that this project did not aim to examine the functionality of these

pluripotency genes but rather to draw conclusions based on the transcriptional profiles we determine; we hope the reviewer agrees that to test functionality of the pluripotency genes by using knockdown studies, lineage tracing and co-binding and ChIP assays at all domains and developmental time points in the ectoderm will take years to do and is beyond the scope of this study. Additionally, as we discuss in the paper, it is very possible that the pluripotency genes also have independent, stage and tissue specific roles, and separating the individual vs shared functions will further add to the complexity of future functional studies. Furthermore, we have added the reference mentioned by the reviewer (*Sci Adv*, 2022; PMID: 35857513), to the existing discussion where we discuss the potential role of the pluripotency genes in guiding lineage commitment.

Per the reviewer's accurate point that we did not perform functional analysis in this work, we have now carefully edited the text to emphasize that all our conclusions are predictions drawn from transcriptional data. Finally, to complement the knockdown studies in figure 5, we have added immunostaining-based quantification of Pax7 in the neural plate border region from embryos that have been individually injected with a translation blocking morpholino to gain insight on the respective functional roles of Nanog, PouV and Klf4 during neural crest development (new Supplemental Fig. 7).

Reviewer #4 (Remarks to the Author):

In this manuscript, Pajanoja and colleagues present a single-cell, spatiotemporal profile of the neural crest (NC). The authors identify that stemness signatures are preserved in the ectoderm and not necessarily restricted to the NC. The manuscript is well written and the breadth of technologies for this study is impressive. In particular, the authors perform multiplexed single cell spatial transcriptomics (scMST), an iterative approach that enables detection of five genes at a time for repeated rounds of hybridisation up to 30 genes. I have some comments regarding the scMST and scRNA-seq components of the study that should be addressed.

- Did the authors assess for any evidence of technical-driven correlations among genes that were selected to be profiled in the same hybridisation round? If so how was this accounted for in downstream analysis?

We thank the reviewer for pinpointing this concern. To maintain the RNA transcripts as diffraction limited individual dots, the HCR-amplification was allowed for only 1.5h, and the signal intensity thus is not saturated, and the levels don't reach extremely high values that would create a risk of the signal bleeding to other channels. We carefully examined all the images and did not observe detection of signal in the neighboring channels, as shown in new Supplementary Fig 1e. Furthermore, the probe sets for each hybridization round were always the same, and the different sets were also always applied to the embryos samples in the same order, and we normalized the genes across each embryo similarly, which helps to minimize any potential technical variations or correlations among the genes/samples. Finally, to confirm the integrity of the RNA throughout the hybridization rounds, the samples were always re-imaged by using the first probe set after the last round, and plotted against the results from the first round (new supplemental figure 1c). In sum, the scMST pipeline is optimized to ensure a fair comparison of gene expression levels of each individual gene within different spatial locations of the individual samples, as well as across the different developmental stages.

- Similarly for individual fields of view, how did the authors ensure that normalisation/thresholding of intensities per colour was adequate, especially for confounding of varying expression patterns spatial

regions with technical fields of view? Figure 1d (right) appears to display some vertical lines which might be driven by field of view effects.

We thank the reviewer for this concern and realize we didn't explain this part of the analysis sufficiently. We have now edited the methods section to explain this better.

We would like to clarify that our analysis methodology does not involve stitching of the individual fields of views; to avoid potential errors or artifacts that may arise from the stitching process such as losing the position of dots (transcripts), the images are stitched after the expression analysis is complete. To enhance the accuracy of our analysis, we apply z-scoring by separately pooling the data from the three fields of views belonging to each embryo. First, we z-score the genes per embryo and then apply another z-scoring across all embryos of the specific stage. We performed rigorous testing of various ways to combine the expression information from the individual fields of views and embryos, and we can confidently state that the analysis method results in a highly accurate distribution of gene scores. Importantly, we can compare the output expression patterns between different embryos and verify that the spatial pattern is similar in all samples. We have now expanded our analysis by incorporating a new approach to z-score gene expression across all stages, allowing for direct comparison of gene expression levels at each developmental time point. In addition to our previous analysis, which focused on stage-specific gene expression and subpopulations of cells within each stage, this new method provides a comprehensive view of gene expression dynamics throughout development. This updated analysis is presented in our new figure (new Figures 2c and d).

- Spatial back-mapping is a visualisation technique where false colour images are created with cells coloured by the cluster to which they have been assigned according to the 30-genes signature. As alluded to in the manuscript, for cells that may be borderline between clusters (e.g. Neural crest stem and Epidermis stem), it would be worth assigning a confidence value to the cluster identity. This could be done by post-hoc supervised learning to extract an associated confidence measure, or alternatively by repeated clustering with a slightly different approach (e.g. gaussian mixture models rather than k-means clustering).

We thank the reviewer for bringing up this concern, which also makes us realize we have not successfully described the method, which may have potentially lead to misunderstandings. To clarify, the hierarchical clustering analysis was done in two dimensions with a cosine similarity between the pairwise objects in the distance matrix. In contrast to our use of K-means clustering to distinguish unspecific probe binding from signal during the signal processing step, we do not use this method to cluster cell populations. Hence, we do not automatically assign cell annotations and cannot provide confidence values for the clusters. To perform cell annotations, we examine the transcriptional profiles of cell subpopulations in the heatmap, in a manner similar to the approach used for scRNAseq data analysis. We have now included further clarification to this in the methods section.

- The authors perform scRNA-seq as a validation of the results obtained from the scMST data. It would be worthwhile directly assessing the concordance between the estimated gene expression profile of scMST and that of scRNA-seq. Additionally, were the authors able to directly integrate these two data together into a joint space? Doing so may enable prediction of additional subclusters or smooth patterns, although this may be limited by the small number of genes profiled and/or the need for z-score normalisation of the scMST data.

We thank the reviewer and appreciate the suggestion of integrating the scMST and scRNAseq data sets. Regrettably, despite our attempts, we must conclude that this task demands advanced coding skills, which are presently beyond our current computational skills. Additionally, as the reviewer states, the clusters

created by the two single cell data sets are not comparable due to the small amount of genes in scMST, and since the scRNAseq clusters are defined by transcriptional similarities across all the 20K genes, it is not possible to identify the same cells in these two data sets. Therefore, we took advantage of the module scoring (Figs 3 and 4) which we used to find the pan-ectodermal stem cells to further analyze the gene expression profile of these cells (vs the others).

- I suggest the authors make the processed scMST data available to readers, either as supplementary or via figshare/zenodo. While the raw images might be large and unwieldy, the cell boundaries, centroids and normalised gene expression values should be quite amenable for public data sharing.

We appreciate the importance of making our data accessible to the scientific community and have now made the processed scMST data available to readers as a supplementary table to the publication. (Supplemental Tables 1, 2, 3 and 4)

REVIEWER COMMENTS

Reviewer #1 (Remarks to the Author):

The response to review and the related revision of the manuscript have addressed/acknowledged that:

- The different stemness/pluripotent-like states of the neural crest cell progenitors were inferred based primarily on gene expression profiles and not the acquisition of lineage propensity.
- The lack of robust validation of the role of the three pluripotency-related factors in the specification of the neural crest cell lineages, other than noting the differential effect of LOF of these factors on Pax7 activity in the NPB domain; and acknowledged that functional analysis is required pending future studies.

Points for further consideration:

a. The data of the expression pattern of the three pluripotency-related factors in hindbrain of the E8 mouse embryo, when compared with data from similar cross-sectional region of the HH7 avian hindbrain may support the notion of a widespread domain of pluripotency-like states across the neural plate. The mouse embryo data do not provide knowledge gain beyond what is known of the pluripotent-like state of the emerging neural crest cells in the mouse. The E8 mouse hindbrain data may be omitted from this report, which would not diminish the merit of the findings in the avian model. This limited scope of findings would not warrant the inference that the “mechanism of NC stemness” is common/conserved in these two vertebrate species. Other aspects of the developmental process, such as the transition through the intermediate state and delimitation of the pluripotent-like lineage-committed state in the NPB, were not elucidated. Regarding the conservation of developmental mechanism, the critical questions of (a) “sustained” or “re-activated” pluripotent-like state, (b) progressive change in pluripotent-like state and lineage commitment and (c) the developmental process is not restricted to axial position in the mouse model have yet to be addressed.

b. It is potentially confusing to designate the multiple types of endocrine cells as “mesodermal” or “endodermal. It may be appropriate to refer the non-ectodermal-like derivatives cell types as those “that normally arise from other germ layers”.

c. Ectodermal pluripotent-like cells: an appropriate term would be “pluripotent-like ectoderm cells”.

d. Pluripotency signature is found in NC at “all axial levels”: it may be noted that only HH13 embryo were studied, and the presumption is that HH13 posterior neural folds are developmentally equivalent to the head region of an HH7-9 embryo. A more robust demonstration would be to examine a developmental series of embryos from HH7-9 to HH13 or beyond to ascertain if the Nanog/Oct4/Klf4 signature is consistently found in the equivalent NPB domain where the NC progenitor cells are at the pre-migratory cell state.

e. The authors clarified that the computational analyses of the various omics data (scMST, scRNA-seq, and bulk RNA-seq) generated for different developmental stages were performed separately. While this may allow high level validation of common key genes captured across different data types, the study falls short from providing a more in-depth and cohesive data resource for future referencing.

f. Regarding the biological replicates and batch effects, it is not clear where in the revised MS that the following information is provided: “Bulk RNaseq samples consist originally of 4 biological replicates (where in each replicate one sample consists of cells from 6-7 embryos), and PCA plots were used to verify that they grouped together, indicating that there is no significant batch effect. The scRNAseq data consists of two biological samples per stage (each sample consisted of cells from 4-7 embryos per sample), and the parallel samples did not present sample specific subclusters, but instead all clusters were represented evenly in both as expected. We have updated the manuscript to make this information more clear.”

g. Supplementary Figure 1a: The clustering of the signal of scMST data was not readily visible. Should not an outlier detection approach be more appropriate if the goal is to identify and retain the true signal in the data. It is not clear why there is a need to partition the data into multiple clusters for distinguishing artifacts.

Reviewer #2 (Remarks to the Author):

I have read the revised manuscript and rebuttal comments in response to points raised in the initial review. The authors have done a good job in rebalancing the paper to make it clear they are focused on analyzing transcriptional signatures and not the functions of pluripotency factors. I feel they have adequately addressed my points and those of the other reviewers. The revised paper reads much better, includes more relevant references and has clarified many of the concerns due to imprecise descriptions. This work makes a significant advance to the field and I support publication.

Reviewer #3 (Remarks to the Author):

I thank the reviewers for their efforts in revising the text, incorporating new data, and responding to the comments. Some further revisions to the text are required to clarify the meaning behind some of the new sections, as the current version is still in some sections hard to follow. I have provided this below with the idea to help communicate better the findings.

The revised title is misleading and is not supported by the experiments performed in the manuscript. The authors have not resolved in this study whether maintenance of pluripotency factors “enables potential for neural crest formation”.

Page 4: “to this day, to define the time at which the NC acquires stemness and whether it is maintained or reactivated, a systematic, multiplexed, high resolution, single-cell level spatiotemporal examination of the stemness characteristics during gastrula- to-neurula stages has not been performed” – the suggested experiment will not address the timing at which the NC acquires stemness. The acquisition of “stemness” requires functional evaluation, (such as examining the identity of a given cell and its daughter cells). This has not been evaluated here so the authors should be careful in their use of the word and ideally, provide clear references that support the definition of “stemness” they are referring to.

Page 5: – “suggesting a widely accessible chromatin state to multiple fates” – it is confusing to read this statement.

Page 8 – “Thus, we focused on Nanog, PouV /Oct4, and Klf4 co-expression to monitor the transcriptional signature of pluripotency due to their known roles in epiblast and embryonic stem cell pluripotency^{29, 30}. We, which from now on will be referred to as pluripotency signature.”

I suggest the authors use “signature of pluripotency-like” from this point onwards, rather than “signature of pluripotency” to help distinguish this particular definition and the remaining passages that refer to this specific definition, which appears to contrast with earlier mentions of the term “signature of pluripotency” in the manuscript (that presumably also include Sox2?).

Page 14 – Can the authors please comment on whether the Klf4 MO injection has altered the size of the neural tube? The MO injected side appears thinner, relative to the control side.

Reviewer #4 (Remarks to the Author):

I thank the authors for their careful work addressing my comments. Particular additions are identifying high positive correlation between first and last rounds of hybridisation, as well as making the processed scMST data available as supplementary table. The manuscript also includes several clarifications that I think will improve the readability of the manuscript and therefore take-up in the research community.

REVIEWER COMMENTS

Reviewer #1 (Remarks to the Author):

The response to review and the related revision of the manuscript have addressed/acknowledged that:

- The different stemness/pluripotent-like states of the neural crest cell progenitors were inferred based primarily on gene expression profiles and not the acquisition of lineage propensity.
- The lack of robust validation of the role of the three pluripotency-related factors in the specification of the neural crest cell lineages, other than noting the differential effect of LOF of these factors on Pax7 activity in the NPB domain; and acknowledged that functional analysis is required pending future studies.

Points for further consideration:

a. The data of the expression pattern of the three pluripotency-related factors in hindbrain of the E8 mouse embryo, when compared with data from similar cross-sectional region of the HH7 avian hindbrain may support the notion of a widespread domain of pluripotency-like states across the neural plate. The mouse embryo data do not provide knowledge gain beyond what is known of the pluripotent-like state of the emerging neural crest cells in the mouse. The E8 mouse hindbrain data may be omitted from this report, which would not diminish the merit of the findings in the avian model. This limited scope of findings would not warrant the inference that the “mechanism of NC stemness” is common/conserved in these two vertebrate species. Other aspects of the developmental process, such as the transition through the intermediate state and delimitation of the pluripotent-like lineage-committed state in the NPB, were not elucidated. Regarding the conservation of developmental mechanism, the critical questions of (a) “sustained” or “re-activated” pluripotent-like state, (b) progressive change in pluripotent-like state and lineage commitment and (c) the developmental process is not restricted to axial position in the mouse model have yet to be addressed.

We thank the reviewer for these additional comments. We examined whether our main finding, the co-expression of pluripotency genes in the entire ectoderm (and not just in the neural crest domain), applies to a mammalian species, and obtained supporting evidence by performing multi-channel FISH to the cranial mouse ectoderm, which indeed showed a similar expression pattern as found in the chick. Thus, to address the reviewer’s current comment, we would like to clarify that the purpose of including the E8 mouse data alongside the HH7 avian data is to

provide additional comparative insights into the expression pattern of pluripotency genes across the two vertebrate species to initiate the discussion of a potentially conserved mechanism. As the reviewer does not raise any criticism towards the quality of the existing mouse data, we do not agree with the suggestion of removing the data. We appreciate the reviewer's point and have now further emphasized in the results that "While our findings provide initial evidence for a conserved mechanism across vertebrates, future studies will be necessary to evaluate the extent of similarity in different model organisms including the mouse".

b. It is potentially confusing to designate the multiple types of endocrine cells as "mesodermal" or "endodermal. It may be appropriate to refer the non-ectodermal-like derivatives cell types as those "that normally arise from other germ layers".

We want to clarify that we have not ever stated or implied that neural crest cell derivatives are endodermal or mesodermal. Instead we have consistently used the terms "endodermal-like" and "mesodermal-like", which we think is accurate in this case to describe phenotypes of the respective derivatives to highlight the exceptionally high stem cell potential of the neural crest cells. We believe that these terms capture the resemblance of these neural crest derived cell types to those that typically arise from the respective other germ layers during embryonic development.

Per the reviewer's suggestion, we have now added the term "non-ectodermal-like" to the introduction. However, we have chosen to retain the terminology "mesodermal-like" and "endodermal-like" in the main text instead of using a more general reference to other germ layers. Our decision to do so is to specifically name the other germ layers, which may assist readers who are not familiar with the details of gastrulation and early embryo development.

c. Ectodermal pluripotent-like cells: an appropriate term would be "pluripotent-like ectoderm cells".

We have changed the term accordingly.

d. Pluripotency signature is found in NC at "all axial levels": it may be noted that only HH13 embryo were studied, and the presumption is that HH13 posterior neural folds are developmentally equivalent to the head region of an HH7-9 embryo. A more robust demonstration would be to examine a developmental series of embryos from HH7-9 to HH13 or beyond to ascertain if the Nanog/Oct4/Klf4 signature is consistently found in the equivalent NPB domain where the NC progenitor cells are at the pre-migratory cell state.

We thank the reviewer for bringing up this point. We agree with the reviewer and have now changed the text to trunk neural crest cells and multiple axial levels instead of “all axial levels” to avoid overstating of our results.

e. The authors clarified that the computational analyses of the various omics data (scMST, scRNA-seq, and bulk RNA-seq) generated for different developmental stages were performed separately. While this may allow high level validation of common key genes captured across different data types, the study falls short from providing a more in-depth and cohesive data resource for future referencing.

The integration of bulk RNAseq, single cell RNAseq, and spatial transcriptomics data poses significant challenges due to their distinct experimental protocols and data structures. Since there are no current means available to combine these data sets (not suggested by the reviewer either) we have analyzed all the approaches separately. We leveraged the unique strengths of each method and further validated our findings through their convergence. While we show multiple commonalities in the results from the different approaches, future will tell how well the provided data serves as a resource for future references.

f. Regarding the biological replicates and batch effects, it is not clear where in the revised MS that the following information is provided: “Bulk RNaseq samples consist originally of 4 biological replicates (where in each replicate one sample consists of cells from 6-7 embryos), and PCA plots were used to verify that they grouped together, indicating that there is no significant batch effect. The scRNAseq data consists of two biological samples per stage (each sample consisted of cells from 4-7 embryos per sample), and the parallel samples did not present sample specific subclusters, but instead all clusters were represented evenly in both as expected. We have updated the manuscript to make this information more clear.”

We thank the reviewer for this point and wish to clarify that the revised manuscript stated the above information in the material and methods section under “chick embryo sample collection”. Additionally, the information regarding the scRNAseq analysis was stated in the main text (in the beginning of the scRNAseq section “scRNAseq supports scMST findings on ectodermal pluripotency-like signature”). The number of parallel bulk RNAseq samples was also visible in the PCA plot in Fig 2e, and we have now added information of the samples in the main text as well (second sentence under the title “ Bulk-RNA-sequencing analysis shows maintenance of pluripotency-like signature in the developing neural crest cells.”). We hope the reviewer now finds our choice of placing the information in the text appropriate.

g. Supplementary Figure 1a: The clustering of the signal of scMST data was not readily visible.

Should not an outlier detection approach be more appropriate if the goal is to identify and retain the true signal in the data. It is not clear why there is a need to partition the data into multiple clusters for distinguishing artifacts.

Per the reviewer's suggestion, to better demonstrate the signal processing step, we have now added the raw image of the example image to be compared with the signal chosen with the k-means clustering in Supplementary Fig 1a .

We appreciate the reviewer's thoughtful consideration of alternative approaches. While outlier detection approaches can be useful in identifying and excluding outliers, we have chosen to employ k-means clustering for several reasons, which we would like to elaborate on:

The primary goal of our signal processing step is to accurately distinguish between diffraction limited dots (RNA transcripts) exhibiting low signal intensities (resulting from unspecific probe binding) and excessively high signal intensities (associated with artifacts) within our spatial transcriptomics data. In our approach, we employ a set of 20 probes per gene whenever possible, taking into consideration that certain genes may pose limitations on the design of 20 probes in their coding sequence (with the lowest number being 13 probes). It is inevitable that some background binding will occur with each probe set. A weak signal is considered background when only a few probes are bound, while a strong signal indicates real binding when all probes are bound. However, the strong signal is probe set dependent, as even in the case of real signal, all the probes are not necessarily always bound, and some genes have more probes and thus a higher signal. Importantly, even within the full set of 20 probes, we cannot determine the exact number of probes that successfully bind, which is also not constant but sample specific. Given the challenges arising from these factors, k-means clustering emerges as an appropriate approach for addressing such complex signal distributions in our data. By partitioning the signal intensities into distinct clusters based on their similarity, k-means clustering enables us to effectively capture the full range of signal variations present in our dataset, including subtle variations that may not be captured by outlier detection approaches. By utilizing an unsupervised learning algorithm, k-means clustering allows us to objectively distinguish different signal intensity levels without prior assumptions or reliance on external reference samples. In our specific experimental context, where clear-cut outliers are absent in the signal intensity distribution as well as well-defined thresholds for outlier detection are lacking, this unbiased approach provided by k-means clustering holds significant value. This also provides a flexible and adaptable framework for partitioning the signal from image to image. Importantly, when each data point (RNA transcript) is assigned to a specific cluster based on its signal intensity, we are able to visualize and understand the spatial distribution of different RNA transcripts within the tissue sample, facilitating further analysis and interpretation.

We have extensively validated our k-means clustering approach by analyzing multiple images and achieving satisfactory results during the pipeline development phase. This validation process involved comparisons with established ground truth images, such as well-known expression patterns of neural crest genes. Our assessments confirmed the effectiveness of k-means clustering in accurately identifying and segregating low and high signal intensities. This successful validation and the compatibility of k-means clustering with our existing analysis pipeline, including downstream analyses and visualization techniques, further strengthened our confidence in its suitability.

We appreciate the reviewer's suggestion regarding the utilization of an outlier detection approach, and we acknowledge the advantages such methods may offer. However, as explained above, in scMST analysis we also exclude weak background signal that does not qualify as outlier signal. We thus firmly believe that the strengths, benefits and ease of use offered by k-means clustering align closely with the goals and requirements of our specific research. We hope this clarification provides a comprehensive understanding of our rationale for selecting and relying on k-means clustering. We have full confidence in the suitability and reliability of our chosen approach for the scMST analysis conducted in this study. For more information, the details of the image analysis steps are provided in the github link.

Reviewer #2 (Remarks to the Author):

I have read the revised manuscript and rebuttal comments in response to points raised in the initial review. The authors have done a good job in rebalancing the paper to make it clear they are focused on analyzing transcriptional signatures and not the functions of pluripotency factors. I feel they have adequately addressed my points and those of the other reviewers. The revised paper reads much better, includes more relevant references and has clarified many of the concerns due to imprecise descriptions. This work makes a significant advance to the field and I support publication.

We are happy that the reviewer is satisfied with our response and sincerely thank for their helpful suggestions that improved our manuscript.

Reviewer #3 (Remarks to the Author):

I thank the reviewers for their efforts in revising the text, incorporating new data, and responding to the comments. Some further revisions to the text are required to clarify the meaning behind some of the new sections, as the current version is still in some sections hard

to follow. I have provided this below with the idea to help communicate better the findings.

The revised title is misleading and is not supported by the experiments performed in the manuscript. The authors have not resolved in this study whether maintenance of pluripotency factors “enables potential for neural crest formation”.

We thank the reviewer for this point and agree that it has been difficult to come up with the most accurate title given the character limit. With the word potential, we aimed to highlight the fact that we don't really study whether some type of neural crest lineage will form even without the pluripotency factors, but realize this may not have come across the way we were hoping. We have now changed the title to “Maintenance of pluripotency-like signature in the entire ectoderm leads to neural crest stem cell potential”. The abstract then clearly states that the study is based on transcriptional profiling analysis.

Page 4: “to this day, to define the time at which the NC acquires stemness and whether it is maintained or reactivated, a systematic, multiplexed, high resolution, single-cell level spatiotemporal examination of the stemness characteristics during gastrula- to-neurula stages has not been performed” – the suggested experiment will not address the timing at which the NC acquires stemness. The acquisition of “stemness” requires functional evaluation, (such as examining the identity of a given cell and its daughter cells). This has not been evaluated here so the authors should be careful in their use of the word and ideally, provide clear references that support the definition of “stemness” they are referring to.

We agree that the word stemness can be interpreted in various ways and is difficult to use. We have now modified the sentence and re-evaluated our terminology throughout the manuscript.

Page 5: – “suggesting a widely accessible chromatin state to multiple fates” – it is confusing to read this statement.

We have removed the chromatin accessibility statement

Page 8 – “Thus, we focused on Nanog, PouV /Oct4, and Klf4 co-expression to monitor the transcriptional signature of pluripotency due to their known roles in epiblast and embryonic stem cell pluripotency^{29, 30}. We, which from now on will be referred to as pluripotency signature.”

I suggest the authors use “signature of pluripotency-like” from this point onwards, rather than “signature of pluripotency” to help distinguish this particular definition and the remaining

passages that refer to this specific definition, which appears to contrast with earlier mentions of the term “signature of pluripotency” in the manuscript (that presumably also include Sox2?).

We have now systematically modified the text and always talk about pluripotency-like signature

Page 14 – Can the authors please comment on whether the Klf4 MO injection has altered the size of the neural tube? The MO injected side appears thinner, relative to the control side.

We thank the reviewer for raising this point. We have now added measurements of the width of the neural tube, and while the KLF4 MO shows a trend towards a reduced width, the difference is not statistically significant, whereas loss of Nanog results in a significantly thinner neural tube (new Supplementary Fig. 7c).

Reviewer #4 (Remarks to the Author):

I thank the authors for their careful work addressing my comments. Particular additions are identifying high positive correlation between first and last rounds of hybridisation, as well as making the processed scMST data available as supplementary table. The manuscript also includes several clarifications that I think will improve the readability of the manuscript and therefore take-up in the research community.

We are happy that the reviewer is satisfied with our response and sincerely thank for their helpful suggestions that improved our manuscript.

REVIEWERS' COMMENTS

Reviewer #1 (Remarks to the Author):

Revisions were made to address critique c, d, f, and g.

Reasonable response to suggestions of further data analysis and experimental work:

- Response a – not extending the mouse embryo study, while qualifying the merit of the finding and indicating the imperatives of follow-up studies.
- Response e – not performing the integrative analysis of various omics data due to technical constraint.

Response b.

As this study was performed on an avian model, it may not be appropriate to consider “the calcitonin secreting cells in the ultimobranchial body” as an example of endodermal-like cells (ref 8: review Perera & Kerosuo 2020).

Suggestion for edits: “non-ectodermal-like cell types that would typically arise from the mesoderm layer” which are “facial bones and cartilage, smooth muscle, (and) adipocytes”.

REVIEWERS' COMMENTS

Reviewer #1 (Remarks to the Author):

Revisions were made to address critique c, d, f, and g.

Reasonable response to suggestions of further data analysis and experimental work:

- Response a – not extending the mouse embryo study, while qualifying the merit of the finding and indicating the imperatives of follow-up studies.
- Response e – not performing the integrative analysis of various omics data due to technical constraint.

Response b.

As this study was performed on an avian model, it may not be appropriate to consider “the calcitonin secreting cells in the ultimobranchial body” as an example of endodermal-like cells (ref 8: review Perera & Kerosuo 2020).

Suggestion for edits: “non-ectodermal-like cell types that would typically arise from the mesoderm layer” which are “facial bones and cartilage, smooth muscle, (and) adipocytes”.

We thank the reviewer for their thorough feedback on our manuscript and are pleased to hear that they are satisfied with the outcome. We wish to clarify that the ultimobranchial body finding is made in the chick embryo, and we want to keep the introduction as it is, describing the evidence of not only mesodermal-like, but also endodermal-like cells derived from the neural crest.